# Overexpression of neuregulin 1 in GABAergic interneurons results in reversible cortical disinhibition

Yao-Yi Wang[1,2], Bing Zhao[1,2], Meng-Meng Wu[1,2], Xiao-Li Zheng[1], Longnian Lin[1] & Dong-Min Yin [1✉]

Cortical disinhibition is a common feature of several neuropsychiatric diseases such as schizophrenia, autism and intellectual disabilities. However, the underlying mechanisms are not fully understood. To mimic increased expression of *Nrg1*, a schizophrenia susceptibility gene in GABAergic interneurons from patients with schizophrenia, we generated gto*Nrg1* mice with overexpression of *Nrg1* in GABAergic interneurons. gto*Nrg1* mice showed cortical disinhibition at the cellular, synaptic, neural network and behavioral levels. We revealed that the intracellular domain of NRG1 interacts with the cytoplasmic loop 1 of Na$_v$1.1, a sodium channel critical for the excitability of GABAergic interneurons, and inhibits Na$_v$ currents. Intriguingly, activation of GABAergic interneurons or restoring NRG1 expression in adulthood could rescue the hyperactivity and impaired social novelty in gto*Nrg1* mice. These results identify mechanisms underlying cortical disinhibition related to schizophrenia and raise the possibility that restoration of NRG1 signaling and GABAergic function is beneficial in certain neuropsychiatric disorders.

[1] Key Laboratory of Brain Functional Genomics, Ministry of Education and Shanghai, School of Life Science, East China Normal University, Shanghai, China. [2] These authors contributed equally: Yao-Yi Wang, Bing Zhao, Meng-Meng Wu. ✉email: dmyin@brain.ecnu.edu.cn

Cortical disinhibition is a common feature of several neuropsychiatric diseases such as schizophrenia, autism, and intellectual disabilities[1]. However, the underlying molecular and cellular mechanisms are not fully understood. Neuregulin 1 (Nrg1) is genetically associated with schizophrenia in diverse populations[2–4]. The Nrg1 gene encodes NRG1 protein, a trophic factor implicated in both neurodevelopment and neurotransmission[5]. NRG1 has over 30 splicing isoforms per the different amino acid sequences in the N-terminus[6]. The C-terminal fragment of NRG1 (NRG1-CTF), which is generated by extracellular cleavage, can be further cleaved by γ-secretase to generate the NRG1-intracellular domain (NRG1-ICD)[6]. The amino acid sequences of NRG1-ICD are more conserved among different species than the N-terminus[7]. Most of the single-nucleotide polymorphisms (SNPs) in the Nrg1 gene that are associated with schizophrenia are localized in intronic, noncoding regions[6], raising the possibility that they may regulate the expression of the Nrg1 gene. While some studies showed isoform 1 alpha of Nrg1 was lower in brains of schizophrenia patients[8,9], other studies reported increased Nrg1 expression or elevated NRG1 signaling in schizophrenia brain[10–14]. Of note, the increase of Nrg1 expression in schizophrenia brain did not correlate with antipsychotic treatment[11,13], suggesting an association with the disorder instead of medication.

Nrg1 is highly expressed in neurons in the mammalian brain[15]. Recent studies based on single-cell RNA sequencing revealed that Nrg1 was expressed in both inhibitory and excitatory neurons from mouse and human prefrontal cotex (PFC)[16,17], a key brain region implicated in schizophrenia[18,19]. Most postmortem studies analyzed the gene expression in the total homogenate of schizophrenia brain, likely masking cell-type-specific alterations due to cellular heterogeneity[20]. Two recent studies reported the transcriptome alterations in GABAegic interneurons (mainly were parvalbumin+) and pyramidal neurons (PN) from the PFC of schizophrenia patients[21,22]. Here, we analyzed the cell-type-specific Nrg1 expression in the GEO database: GSE93577 and GSE93987[21,22]. We found that Nrg1 expression was increased in GABAergic interneurons but not in PN from the PFC of schizophrenia patients, compared with age- and sex-matched healthy controls.

To mimic increased Nrg1 expression in GABAergic interneurons from schizophrenia patients, we generated gtoNrg1 mice in which Nrg1 overexpression specifically occurred in GABAergic interneurons and could be turned off by doxycycline (Dox). Intriguingly, gtoNrg1 mice showed cortical disinhibition at the cellular, synaptic, and neural network levels. The mechanistic study suggested that NRG1-ICD could interact with Na$_v$1.1, a sodium channel critical for the excitability of GABAergic interneurons and inhibit the Na$_v$ currents. We further demonstrated that cortical disinhibition led to behavioral deficits in gtoNrg1 mice. Lastly, both cortical disinhibition and behavioral deficits in gtoNrg1 mice disappeared when NRG1 expression returned to normal in adulthood. Together, our results demonstrate mechanisms underlying cortical disinhibition related to schizophrenia and shed light on the pathophysiology of neuropsychiatric disorders.

## Results

### Generation of gtoNrg1 mice to mimic increased Nrg1 expression in GABAergic interneurons from schizophrenia patients.
To address the cell-type-specific alteration of Nrg1 gene expression in the postmortem PFC of schizophrenia patients, we analyzed the GEO database GSE93577[21] and GSE93987[22]. The database GSE93577 was composed of gene expression data in GABAergic interneurons from 36 schizophrenia patients and 36 matched healthy controls (Supplementary Table 1). The database GSE93987 included the gene expression data in layer 3 PN from the same PFC samples as database GSE93577 (Supplementary Table 1). Type I Nrg1 mRNA levels were significantly increased in GABAergic interneurons from schizophrenia PFC, compared with age- and sex-matched controls (Fig. 1a). By contrast, Nrg1 mRNA levels were not significantly altered in PN from schizophrenia PFC, compared to age- and sex-matched controls (Fig. 1b). The increased type I Nrg1 expression in GABAergic interneurons from schizophrenia PFC seems not to be due to the antipsychotic treatment because type I Nrg1 mRNA levels were similar between drug-naive and drug-treated schizophrenia patients (Supplementary Fig. 1a). There is no sex difference for type I Nrg1 expression in the schizophrenia PFC (Supplementary Fig. 1b). Type IV Nrg1 expression was not significantly increased in GABAergic interneurons from schizophrenia PFC (Supplementary Fig. 1c), suggesting a specificity for an increase of type I Nrg1 expression in GABAergic interneurons. Neuregulin 3 (Nrg3) is another schizophrenia risk factor among the neuregulin gene family[23–25]. By contrast, the mRNA levels of Nrg3 were reduced in GABAergic interneurons from schizophrenia PFC, compared with age- and sex-matched controls (Supplementary Fig. 1d). Together, these results suggest that type I Nrg1 expression is increased in GABAergic interneurons from the postmortem PFC of schizophrenia patients.

To mimic the increased type I Nrg1 expression in GABAergic interneurons from schizophrenia patients, we aim to generate Nrg1 transgenic mice that overexpress type I Nrg1 specifically in GABAergic interneurons. Toward this goal, TRE-Nrg1 mice (Fig. 1c)[26] were crossed with Gad67-tTA mice which express tTA specifically in GABAergic interneurons[27] (Supplementary Fig. 2). Resulting Gad67-tTA; TRE-Nrg1 mice (gtoNrg1 for Gad67 promoter-driven tet-off Nrg1) produced HA-NRG1 in GABAergic interneurons in the absence of Dox (Fig. 1d). We used Gad67-tTA mice as controls in the following experiments (see also the methods). As shown in Fig. 1e, the mRNA of Nrg1 was increased in several brain regions including olfactory bulb (OB), striatum (STR), and prefrontal cortex (PFC) in gtoNrg1 mice, compared with controls. The increase was due to the expression of the transgene, which was detectable by the expression of the HA tag (Fig. 1f). In agreement, the protein levels of NRG1 (including the full length and ICD) were increased by 50–100% in the PFC of gtoNrg1 mice (Fig. 1g, h). To demonstrate the Nrg1 transgene is specifically expressed in GABAergic interneurons in the PFC, we performed single-cell RT-PCR. As shown in Supplementary Fig. 1e, the Nrg1 transgene indicated by the expression of HA tag was expressed in GABAergic interneurons but not in PN. Lastly, overexpression of Nrg1 could be switched off by feeding the mice with Dox-containing water (Fig. 1e–h). Together, the results indicate that gtoNrg1 mice express higher levels of Nrg1 specifically in GABAergic interneurons, and the overexpression could be turned off efficiently by Dox.

### Behavioral deficits in gtoNrg1 mice.
gtoNrg1 mice showed similar body weight with control littermates (Supplementary Fig. 3a), indicating no growth retardation. However, they were hyperactive in the open-field test (Fig. 2a), a phenotype related to abnormalities of brain dopamine levels in schizophrenia patients. By contrast, the staying time in the margin and center of the open filed was similar between control and gtoNrg1 mice (Supplementary Fig. 3b), indicating no anxiety-like phenotype. Prepulse inhibition (PPI) is a common test of sensorimotor gating that is often impaired in schizophrenia patients[28]. gtoNrg1 mice had a normal response to 75–85 dB background noise (Supplementary Fig. 3c–e). The prepulse itself has significant effects on PPI in both control and

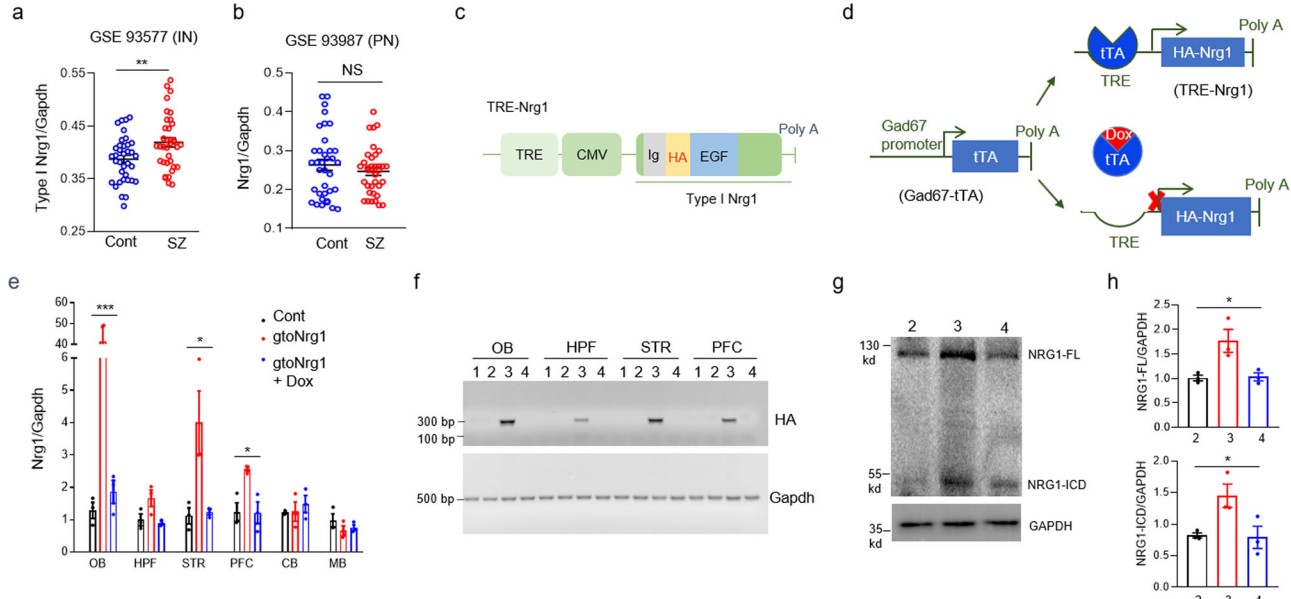

**Fig. 1 Generation of gtoNrg1 mice to mimic increased Nrg1 expression in GABAergic interneurons from schizophrenia patients. a** Increased type I *Nrg1* mRNA levels in GABAergic interneurons from schizophrenia PFC, compared with age- and sex-matched controls. **\*\*P = 0.0064, two-sided *t* test, *n* = 36 for each group. Data are presented as mean values +/− SEM. **b** Similar *Nrg1* expression in pyramidal neurons from layer 3 PFC between control and schizophrenia patients. NS, not significant, two-sided *t* test, *n* = 36 for each group. Data are presented as mean values +/− SEM. Cont control, SZ schizophrenia, IN GABAergic interneurons, PN pyramidal neurons. The levels of *Nrg1* mRNA were normalized to that of Gapdh. **c** Full-length *Nrg1* type Iβ1a was cloned in pMM400 between the promoter complex of TRE and CMV (cytomegalovirus minimal promoter) and SV40 polyadenylation (Poly A) signal. An HA tag was inserted between the Ig and EGF domain of NRG1. **d** The principle of the tet-off system. HA-Nrg1 is expressed in *Gad67*-tTA; TRE-*Nrg1* (gto*Nrg1*) mice. The expression of HA-Nrg1 can be switched off by Dox. **e** Increased *Nrg1* mRNA levels in different brain regions of gto*Nrg1* mice. The *Nrg1* mRNA levels return to normal after Dox treatment for 5 days. *P (PFC) = 0.0157, *P (STR) = 0.0223, ***P = 0.001, one-way-ANOVA, *n* = 3 mice for each group. Data are presented as mean values +/− SEM. OB olfactory bulb, HPF hippocampus formation, STR striatum, PFC prefrontal cortex, CB cerebellum, MB midbrain. **f** Expression of HA tag in gto*Nrg1* mice. Three independent experiments were repeated to get similar results. The homogenate from different brain regions was subjected to RT-PCR assay for HA and Gapdh. (1) TRE-*Nrg1*; (2) *Gad67*-tTA; (3) gto*Nrg1*; (4) gto*Nrg1* + Dox. **g** Increased NRG1 protein levels (including the full length and ICD) in the PFC of gto*Nrg1* mice. The homogenate of PFC from different mice was subjected to western blot and probed with anti-NRG1 and anti-GAPDH Abs. (2) control; (3) gto*Nrg1*; (4) gto*Nrg1* + Dox. **h** Quantification of the expression of full-length NRG1 (top) and NRG1-ICD (bottom) in panel **g**. *P = 0.0178 for NRG1-FL, *P = 0.034 for NRG1-ICD, one-way-ANOVA, *n* = 3 mice for each group, data were normalized to controls. Data are presented as mean values +/− SEM.

gto*Nrg1* mice (Supplementary Fig. 3f, g). However, the startle response to 120 dB and PPI were compromised in gto*Nrg1* mice compared with controls (Fig. 2b, c). In the prepulse + pulse trails, both genotypes reduce their startle response with increasing intensity of the prepulse (Fig. 2d). There is a minor but significant difference in the extent to which the prepulse gates the startle response to the pulse between two genotypes (interaction (prepulse × genotype) $F (2, 84) = 3.136, P = 0.0486$, two-way ANOVA) (Fig. 2d). Together, these results suggest that overexpression of *Nrg1* in GABAergic interneurons impairs startle response and has minor effects on sensorimotor gating.

Impaired social behavior is a negative symptom of schizophrenia[29]. gto*Nrg1* mice could distinguish the stimulus mouse and the objective in the three-chamber test, similar to controls (Supplementary Fig. 3h, i), suggesting normal social interaction. However, gto*Nrg1* mice spent lesser time in the chamber with novel mice (S2) but stayed longer in the chamber with familiar mice (S1), compared with controls (Fig. 2e–g), indicating impaired social novelty. The reduced social novelty of gto*Nrg1* mice may not result from the deficient olfaction because gto*Nrg1* mice showed better olfaction during finding the buried food compared with controls (Supplementary Fig. 3j). We also studied nesting behavior in control and gto*Nrg1* mice to assess their ability to establish an organized behavior. Compared with controls, gto*Nrg1* mice were unable to build an identifiable nest within 12 h (Fig. 2h, i).

We next determined whether overexpressing *Nrg1* in GABAergic interneurons affected spatial recognition memory in Y maze. To exclude the potential influence of hyperactivity in the Y-maze test, we analyzed the time exploring the start, old and new arms (Fig. 2j, k). Although there was a preference for the new arm in both control and gto*Nrg1* mice (Fig. 2l, m), the percentage of time exploring the new arm was significantly reduced in gto*Nrg1* mice compared with controls (Fig. 2n). These results suggest that gto*Nrg1* mice have impaired spatial recognition memory.

**Hypersynchrony of neural network in the PFC of gtoNrg1 mice.** Given the evidence of PFC-mediated behavioral deficit such as an impaired social novelty in gto*Nrg1* mice, we next studied how the neural network of PFC was altered in gto*Nrg1* mice. Here, we focus on layers 2–3 of PFC because the layer 3 circuitry in PFC is significantly altered in schizophrenia patients[30]. To this end, we carried out local field potential (LFP) recordings in the PFC of freely behavioral control and gto*Nrg1* mice. We acutely implanted 32-channel tetrodes in the layer 2–3 of PrL (prelimbic cortex, a major region of PFC, Fig. 3a), and after 2 weeks of recovery, the LFP was recorded. We analyzed epochs of activity in which the speed of movement was above 3 cm/s to reduce variability in LFP recordings, and we verified that the mean speed of the epochs analyzed was similar for both genotypes. Analysis of spontaneous LFPs in layers 2–3 of PrL revealed a significant

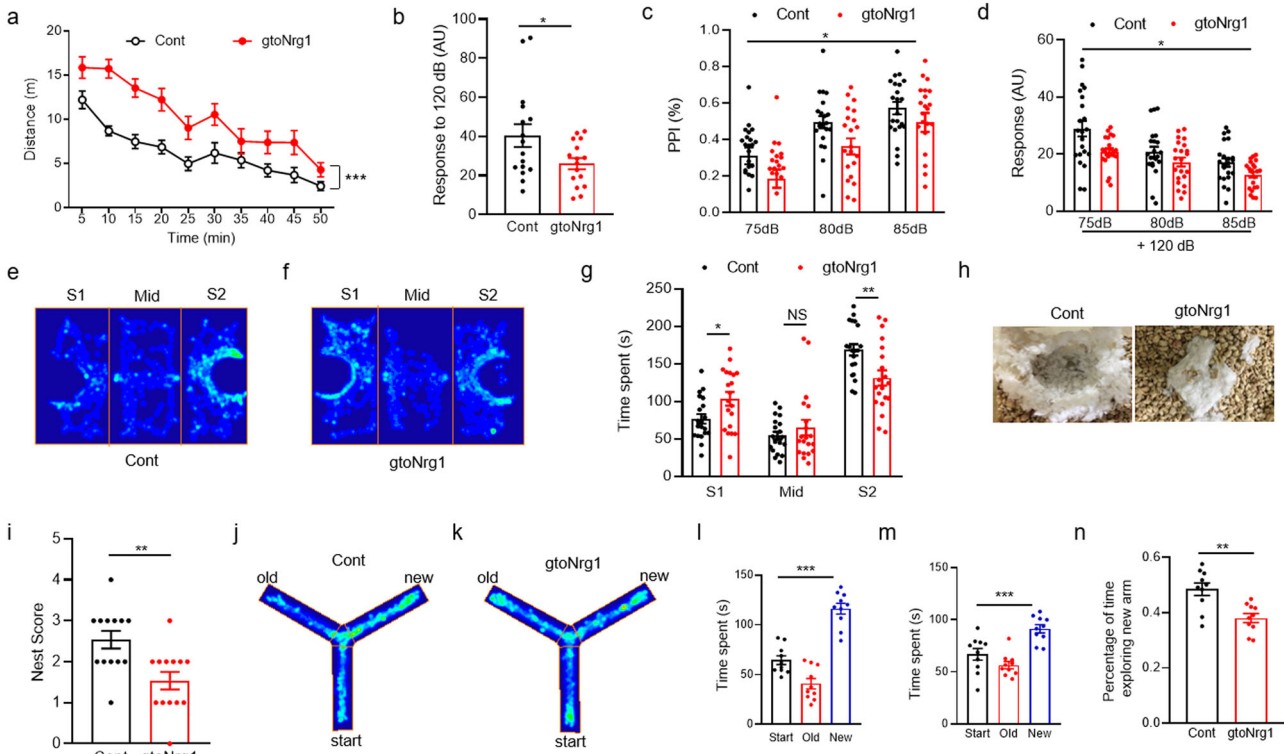

**Fig. 2 Behavioral deficits in gto*Nrg1* mice. a** Travel distance in open field was increased in gto*Nrg1* mice compared with controls. ***Genotype $F(1, 28) =$ 15.74, $P = 0.0005$, two-way ANOVA, $n = 15$ for each group. Data are presented as mean values $+/-$ SEM. **b** Reduced startle response to 120 dB noise in gto*Nrg1* mice compared with controls. *$P = 0.0356$, two-sided $t$ test, $n = 16$ for each group. Data are presented as mean values $+/-$ SEM. **c** Impaired PPI in gto*Nrg1* mice compared with controls. *Genotype $F(1, 42) = 4.9$, $P = 0.0324$, two-way ANOVA, $n = 22$ for each group. Data are presented as mean values $+/-$ SEM. **d** Reduced reflex amplitudes on prepulse + pulse trials in gto*Nrg1* mice compared with controls. *Genotype $F(1, 42) = 6.11$, $P = 0.0176$, ***Prepulse $F(1.709, 71.78) = 53.69$, $P < 0.0001$, *Interaction (prepulse × genotype) $F(2, 84) = 3.136$, $P = 0.0486$, two-way ANOVA, $n = 22$ for each group. Data are presented as mean values $+/-$ SEM. **e, f** Occupancy plot of the heads from control (**e**) and gto*Nrg1* (**f**) mice in the three-chamber test. S1, familiar mouse; S2, novel mouse. **g** Reduced social novelty in gto*Nrg1* mice compared with controls. Time spent in each chamber was quantified. NS not significant, *$P = 0.02$, **$P = 0.0056$, two-sided $t$ test, $n = 20$ for each group. Data are presented as mean values $+/-$ SEM. **h, i** Impaired nest building in gto*Nrg1* mice compared with controls. **h** Representative images for a nest built after 12 h. **i** Quantification of nest score. **$P = 0.0031$, two-sided $t$ test, $n = 13$ for each group. Data are presented as mean values $+/-$ SEM. **j, k** Occupancy plot of the heads from control (**j**) and gto*Nrg1* (**k**) mice in Y maze. **l, m** Preference for the new arm in control (**l**) and gto*Nrg1* (**m**) mice. The time exploring the start, old, and new arms was quantified. ***$P < 0.001$, one-way ANOVA, $n = 10$ for each group. Data are presented as mean values $+/-$ SEM. **n** Impaired spatial recognition memory in gto*Nrg1* mice compared with controls. The percentage of time exploring a new arm in the Y maze was quantified. **$P = 0.0016$, two-sided $t$ test, $n = 10$ for each group. Data are presented as mean values $+/-$ SEM.

increase of activity in gto*Nrg1* mice, compared with controls (Fig. 3b, c). Next, we analyzed the relative power of oscillations in control and gto*Nrg1* mice. We observed a significant increase in the relative power of delta (0.5–3 Hz), theta (4–12 Hz), alpha (13–15 Hz), and beta (16–30 Hz) oscillations in gto*Nrg1* mice, compared with controls (Fig. 3d–k). The relative power of gamma (30–90 Hz) and high-frequency oscillation (HFO) (>100 Hz) were not significantly altered in the PFC of gto*Nrg1* mice, compared to controls (Fig. 3f, l–n). These results suggest that overexpression of *Nrg1* in GABAergic interneurons led to hypersynchrony of the neural network in the PFC.

**Elevated E/I balance in the PFC of gto*Nrg1* mice.** The hypersynchrony of neural networks in gto*Nrg1* mice may reflect an alteration in E/I balance. Next, we sought to determine whether the E/I ratio was changed in the PFC of gto*Nrg1* mice. Toward this aim, we performed a whole-cell patch clamp to record the spontaneous excitatory and inhibitory postsynaptic currents (sEPSC and sIPSC) from layers 2 to 3 PN in PrL (Fig. 4a). As shown in Fig. 4b–f, the frequency but not the amplitude of sEPSC was significantly increased in gto*Nrg1* mice, compared with

controls. By contrast, the frequency but not amplitude of sIPSC was reduced in gto*Nrg1* mice, compared with controls (Fig. 4g–k). In support of these results was the observation that the E/I ratio was elevated in the gto*Nrg1* PFC, compared with controls (Fig. 4l). The reduced sIPSC frequency in gto*Nrg1* mice (Fig. 5g, h, j) might be due to the decreased GABAergic synapse number or the impaired ability of GABA release. However, these possibilities could be small because the miniature IPSC (mIPSC) (Supplementary Fig. 4a–c) and paired-pulse ratio (PPR) of evoked IPSC (eIPSC) (Supplementary Fig. 4d, e) from layers 2 to 3 PN in the PrL were similar between control and gto*Nrg1* mice.

We next investigate whether the reduced sIPSC frequency in gto*Nrg1* mice results from the lower excitability of GABAergic interneurons. To visualize the GABAergic interneurons, we crossed *Gad67*-tTA and gto*Nrg1* mice with TRE-Histone 2B (H2B)-GFP mice[31] to get gtoGfp and gto*Nrg1*; Gfp mice, respectively. These mice express H2B-GFP specifically in GABAergic interneurons (Supplementary Fig. 2) and were subject to whole-cell recording in PFC slices (Fig. 4m). The gto*Nrg1*; Gfp mice showed normal laminar structure and densities of GABAergic interneurons in PFC (Supplementary Fig. 5). The

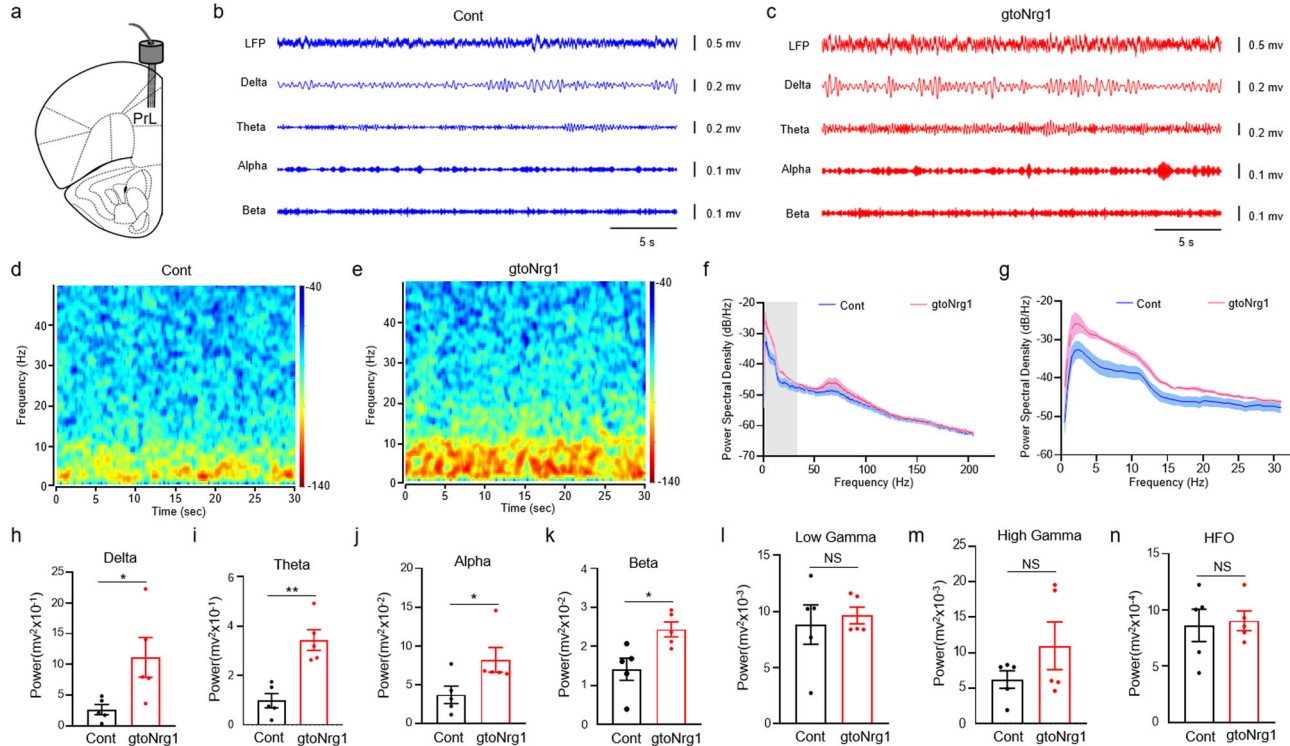

**Fig. 3 Increased delta, theta, alpha, and beta oscillation in the PFC of freely behavioral gto*Nrg1* mice. a** Diagram to show the LFP recording in layers 2–3 of PrL. **b**, **c** Spontaneous LFP and filtered delta, theta, alpha, and beta component of the signal from control (**b**) and gto*Nrg1* mice (**c**). **d**, **e** Power spectrogram of LFP from control (**d**) and gto*Nrg1* mice (**e**). **f** Power spectral density of LFP from 0.5 to 200 Hz in control and gto*Nrg1* mice. $n = 5$ for each group. Data are presented as mean values $+/-$ SEM. **g** Power spectral density of LFP from 0.5 to 30 Hz in control and gto*Nrg1* mice. $n = 5$ for each group. Data are presented as mean values $+/-$ SEM. **h**–**n** LFP band-power in the delta (0.5–3 Hz) (**h**), theta (4–12 Hz) (**i**), alpha (13–15 Hz) (**j**), beta (16–30 Hz) (**k**), low gamma (30–50 Hz) (**l**), high gamma (55–90 Hz) (**m**), and high-frequency oscillation (HFO, 100–300 Hz) (**n**). NS, not significant, *$P = 0.0345$ for panel **h**, **$P = 0.0013$ for panel **i**, *$P = 0.0498$ for panel **j**, *$P = 0.0174$ for panel k, two-sided $t$ test, $n = 5$ for each group. Data are presented as mean values $+/-$ SEM.

firing rate varied between fast-spiking (FS) and non-FS-GABAergic interneurons. Here, we focus on FS-GABAergic interneurons whose function is critical for neuronal synchronization and is heavily diminished in the PFC from schizophrenia patients[32]. Intriguingly, the FS-GABAergic interneurons showed a downward shift of input–output (I/O) curves of action potential (AP) in gto*Nrg1*; Gfp mice, compared with gtoGfp mice (Fig. 4n, o), suggesting reduced excitability of FS-GABAergic interneurons in gto*Nrg1*; Gfp mice.

To further study whether the disinhibition of FS-GABAergic interneurons would increase the firing rate of PN, we recorded AP from PN in control and gto*Nrg1* PFC slices (Fig. 4p). As shown in Fig. 4q, r, the PN firing was significantly increased in gto*Nrg1* mice compared with controls. However, the electrophysiological characteristics of PN were not altered in gto*Nrg1* mice (Supplementary Table 2), indicating that the increased firing rate of PN was secondary to GABAergic hypofunction. Taken together, these results demonstrate that overexpressing NRG1 in GABAergic interneurons causes GABAergic deficit and increases the E/I ratio.

**Reduced Na$_v$ currents in GABAergic interneurons from gto*Nrg1* mice.** The reduced excitability of FS-GABAergic interneurons in gto*Nrg1*; Gfp mice was reflected by the higher rheobase, the minimal currents to produce an AP (Fig. 5a). The intrinsic excitability of GABAergic interneurons was determined by several factors, such as rest membrane potential (RMP), action potential threshold (APT), and afterhyperpolarization (AHP). To isolate intrinsic AP waveform

characteristics, we evoked a single AP with a brief suprathreshold current injection. As shown in representative traces and corresponding dV/dt phase plots (Fig. 5b), the FS-GABAergic interneurons from gto*Nrg1*; Gfp mice had a depolarized APT, compared with gtoGfp mice. In accord, APT was significantly increased in FS-GABAergic interneurons from gto*Nrg1*; Gfp mice, compared with gtoGfp mice (Fig. 5a), indicating a role of depolarized APT in reduced excitability. Voltage-gated sodium (Na$_v$) channels play critical roles in AP generation and threshold[33]. To address whether *Nrg1* overexpression affected Na$_v$ currents, we performed voltage-clamp recording on GABAergic interneurons from gtoGfp and gto*Nrg1*; Gfp PFC slices (Fig. 5c). The I/V curves of Na$_v$ channel activation in GABAergic interneurons showed a downward shift in gto*Nrg1*; Gfp mice, compared with gtoGfp mice (Fig. 5d). In accord, the maximal Na$^+$ current density was significantly decreased in GABAergic interneurons from gto*Nrg1*; Gfp mice, compared with gtoGfp mice (gto*Nrg1*; Gfp: $-185.85 \pm 13.97$ pA/pF, $n = 13$, GtoGfp: $-267.63 \pm 22.41$ pA/pF, $n = 17$, $P = 0.0077$, $t$ test). By contrast, membrane depolarizations required for half-maximal activation of Na$_v$ channels in GABAergic interneurons were similar between gto*Nrg1*; Gfp and gtoGfp mice ($V_{1/2} = -29.86 \pm 0.76$ mV for gto*Nrg1*; Gfp, $n = 13$, $V_{1/2} = -29.32 \pm 0.56$ mV for gtoGfp, $n = 17$, $P = 0.9257$, $t$ test) (Fig. 5e). Together, these results indicated that *Nrg1* overexpression attenuated the peak Na$^+$ current density but without affecting the voltage dependence of Na$_v$ channel activation.

The reduced Na$^+$ current density from gto*Nrg1*; Gfp mice could be due to the lower protein levels or the impaired function

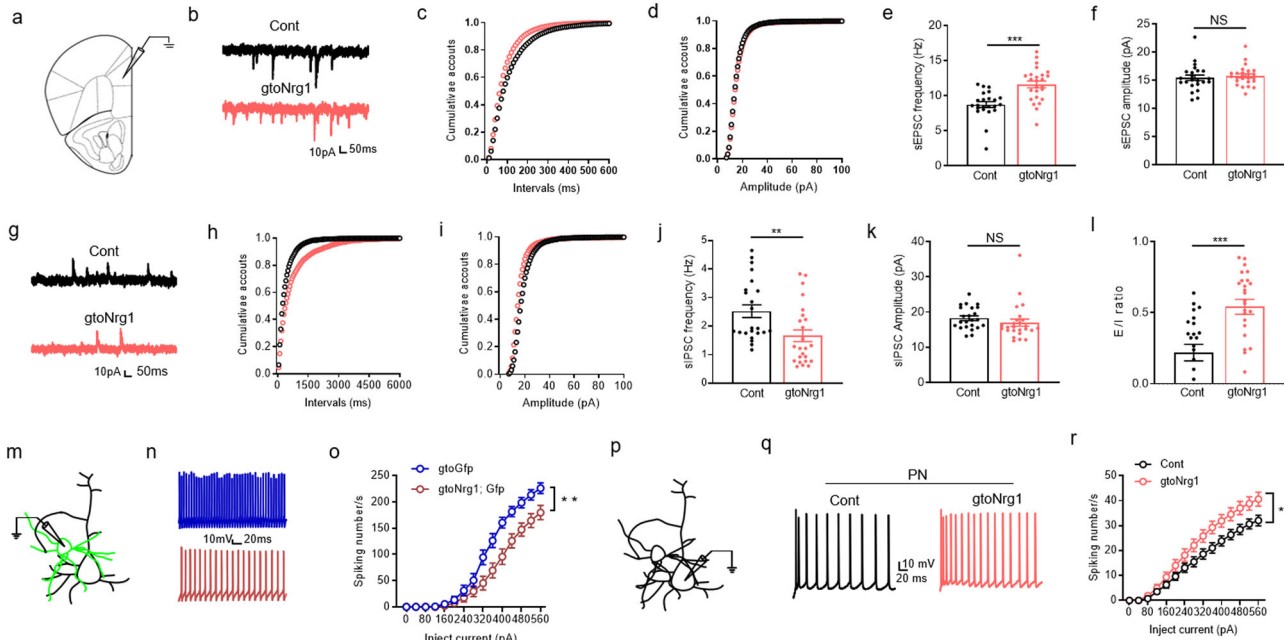

**Fig. 4 Elevated E/I balance in the PFC of gto*Nrg1* mice. a** Diagram to show the whole-cell recording in layers 2–3 of PrL. **b** Representative traces of sEPSC from control and gto*Nrg1* mice. **c, d** Cumulative plots of sEPSC frequency (**c**) and amplitude (**d**). **e** Increased sEPSC frequency in gto*Nrg1* mice. ***$P <$ 0.0001, two-sided $t$ test, $n = 23$ cells from four control mice, $n = 25$ cells from four gto*Nrg1* mice. Data are presented as mean values $+/-$ SEM. **f** Similar sEPSC amplitude between control and gto*Nrg1* mice. NS not significant, two-sided $t$ test, $n = 23$ cells from four control mice, $n = 25$ cells from four gto*Nrg1* mice. Data are presented as mean values $+/-$ SEM. **g** Representative traces of sIPSC from control and gto*Nrg1* mice. **h, i** Cumulative plots of sIPSC frequency (**h**) and amplitude (**i**). **j** Reduced sIPSC frequency in gto*Nrg1* mice. **$P = 0.0068$, two-sided $t$ test, $n = 23$ cells from four control mice, $n = 25$ cells from four gto*Nrg1* mice. Data are presented as mean values $+/-$ SEM. **k** Similar sIPSC amplitude between control and gto*Nrg1* mice. NS not significant, two-sided $t$ test, $n = 23$ cells from four control mice, $n = 25$ cells from four gto*Nrg1* mice. Data are presented as mean values $+/-$ SEM. **l** Elevated E/I ratio in gto*Nrg1* mice. ***$P = 0.0002$, two-sided $t$ test, $n = 23$ cells from four control mice, $n = 25$ cells from four gto*Nrg1* mice. Data are presented as mean values $+/-$ SEM. **m** Diagram to show the recording of GABAergic interneurons expressing EGFP. **n** Representative action potentials of FS-GABAergic interneurons. **o** Reduced excitability of FS-GABAergic interneurons in gto*Nrg1*; Gfp mice, compared with gtoGfp mice. Shown are the I/O curves of action potentials from FS-GABAergic interneurons. **Genotype $F_{(1, 36)} = 8.813$, $P = 0.0053$, two-way ANOVA, $n = 17$ cells from five gtoGfp mice, $n = 21$ cells from four gto*Nrg1*; Gfp mice. Data are presented as mean values $+/-$ SEM. **p** Diagram to show the recording of PN. **q** Representative action potentials of PN. **r** Increased excitability of PN in gto*Nrg1* mice compared with controls. Shown are the I/O curves of action potentials from PN. *Genotype $F_{(1, 50)} = 5.041$, $P = 0.0292$, two-way ANOVA, $n = 27$ cells from five control mice, $n = 25$ cells from four gto*Nrg1* mice. Data are presented as mean values $+/-$ SEM.

of Na$_v$ channels caused by *Nrg1* overexpression. There are four isoforms of Na$_v$ channels (Na$_v$ 1.1, Na$_v$ 1.2, Na$_v$ 1.3, and Na$_v$ 1.6) which are primarily expressed in the central nervous system[33]. The α-subunits of Na$_v$ are necessary for forming a functional ion-selective channel. The α-subunits of Na$_v$ 1.1, Na$_v$ 1.2, Na$_v$ 1.3, and Na$_v$ 1.6 are encoded by the gene Scn1a, Scn2a1, Scn3a, and Scn8a. We analyzed the transcription levels of the four Scn genes in the seven major neuronal clusters from the mouse frontal cortex accessible through an online database DropViz (https://dropviz. org)[16] (Supplementary Fig. 6). Scn1a, Scn2a1, and Scn8a are the three major Scn isoforms expressed in GABAergic interneurons (Fig. 5f). The SCN1A, SCN2A1, and SCN8A proteins are similar in structure and have 74% similarity in the overall amino acid sequences[33]. The expression levels of Scn1a was higher in GABAergic interneurons, but lower in PN compared with Scn2a1 and Scn8a (Fig. 5f, g). These results are consistent with the previous finding that Scn1a is highly expressed in PV-positive GABAergic interneurons and is critical for their excitability[34]. Due to these reasons, we focused on Scn1a in the following study. Neither the mRNA nor the protein levels of Scn1a were reduced in gto*Nrg1* PFC, compared to controls (Supplementary Fig. 7a, b). The SCN1A protein levels in the membrane fraction were also similar between control and gto*Nrg1* PFC (Supplementary

Fig. 7c). Together with the data in Fig. 5a–e, these results suggest that overexpression of *Nrg1* impairs the function of Na$_v$ channels rather than inhibits the gene expression or membrane trafficking of SCN1A.

**Inhibition of Na$_v$ currents in GABAergic interneurons by NRG1-ICD.** ErbB4 is the main receptor of NRG1 in GABAergic interneurons[35–37]. We next determine whether the protein levels or activity of ErbB4 were changed in gto*Nrg1* mice. To this end, the homogenate of PFC from control and gto*Nrg1* mice were subjected to western blots and probed with antibodies against ErbB4 and p-ErbB4. As shown in Supplementary Fig. 8, the protein levels of ErbB4 and p-ErbB4 were not altered in gto*Nrg1* PFC, compared to controls. These results suggest that ErbB4 was not activated in the PFC of gto*Nrg1* mice. To study whether the NRG1 EGF domain could reduce Na$_v$ currents in GABAergic interneurons, we incubated the PFC slices with 5 nM NRG1 EGF domain or BSA (as a control). NRG1 EGF domain could promote evoked GABA release (Supplementary Fig. 9a, b), which is consistent with our previous findings[38–40]. However, the NRG1 EGF domain did not alter the Na$_v$ currents in GABAergic interneurons (Supplementary Fig. 9c, d).

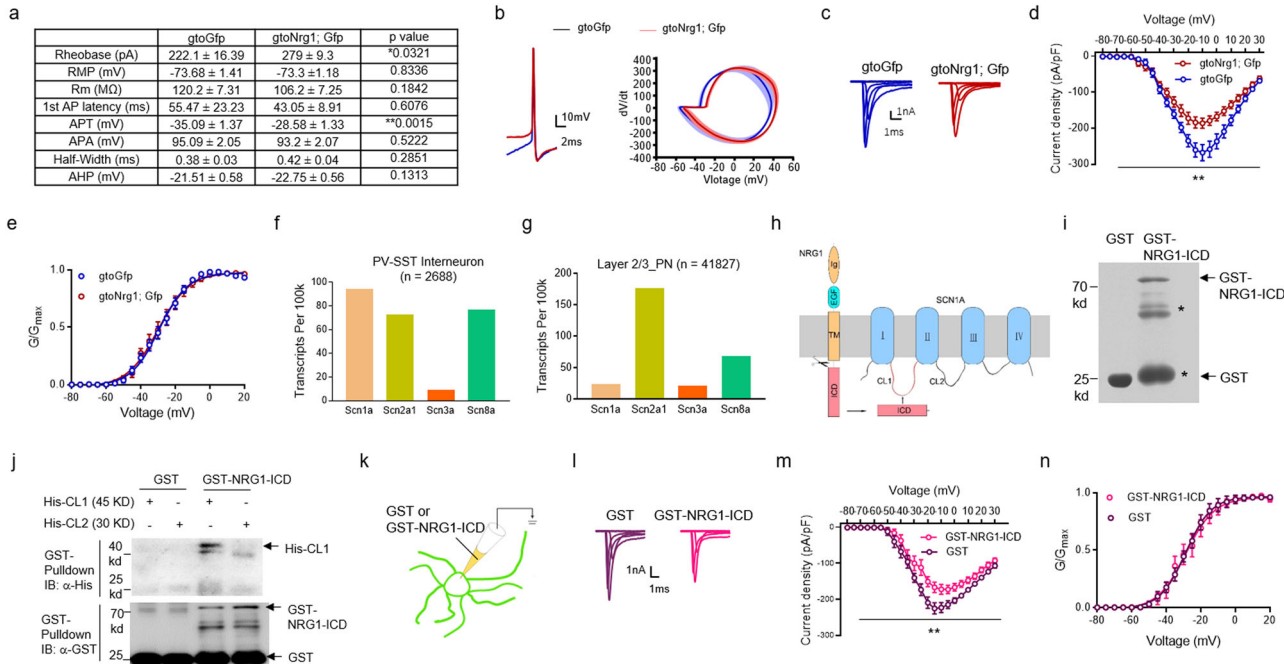

**Fig. 5 Inhibition of Na$_v$ currents in GABAergic interneurons by NRG1-ICD. a** Increased rheobase and depolarized action potential threshold (APT) in the FS-GABAergic interneurons of gto*Nrg1*; Gfp mice, compared with gtoGfp mice. *$P = 0.0321$, **$P = 0.0015$, two-sided $t$ test, $n = 17$ cells from five gtoGfp mice, $n = 21$ cells from four gto*Nrg1*; Gfp mice. **b** Representative traces of a single AP evoked from a suprathreshold current injection (left) and corresponding phase plots (dV/dt vs voltage) (right) recorded in FS-GABAergic interneurons from gtoGfp and gto*Nrg1*; Gfp mice. **c** Reduced Na$^+$ current density in GABAergic interneurons from gto*Nrg1*; Gfp PFC. Representative current traces of Na$_v$ channels in GABAergic interneurons from gto*Nrg1*; Gfp and gtoGfp mice. Currents were elicited by step depolarizations from $-80$ to $+30$ mV in 5 mV increments from a holding potential of $-80$ mV. The tracts shown are for depolarizations from $-80$ to $+20$ mV. **d** I/V curves of Na$_v$ channel activation in GABAergic interneurons from gto*Nrg1*; Gfp and gtoGfp mice. **Genotype $F_{(1, 28)} = 8.264$, $P = 0.0076$, two-way ANOVA, $n = 13$ cells from three gto*Nrg1*; Gfp mice, $n = 17$ cells from four gtoGfp mice. Data are presented as mean values $+/-$ SEM. **e** Similar voltage-dependent activation curves of Na$_v$ channels in GABAergic interneurons from gto*Nrg1*; Gfp and gtoGfp mice. Genotype $F_{(1, 28)} = 0.023$, $P = 0.879$, two-way ANOVA, $n = 13$ cells from three gto*Nrg1*; Gfp mice, $n = 17$ cells from four gtoGfp mice. Data are presented as mean values $+/-$ SEM. **f** Transcriptional levels of Scn1a is higher than Scn2a1, Scn3a, and Scn8a in PV and SST-positive GABAergic interneurons ($n = 2688$ cells). Shown are transcripts of Scn genes per 100 k total transcripts from single-cell RNA sequencing. **g** Transcriptional levels of Scn1a were lower than Scn2a1 and Scn8a in layer 2/3 PN ($n = 41827$ cells). Shown are transcripts of Scn genes per 100 k total transcripts from single-cell RNA sequencing. **h** Diagram showing the structure of SCN1A. The SCN1A protein was composed of four transmembrane domains (I–IV) and two major cytoplasmic loops (CL1 and CL2). The NRG1-ICD could interact with the CL1 of SCN1A. **i** Coomassie blue staining of 30 μg GST and GST-NRG1-ICD proteins. Asterisks indicated degradation product of GST-NRG1-ICD proteins. Four independent experiments were repeated to get similar results. **j** Interaction of NRG1-ICD with His-CL1. The recombinant GST-NRG1-ICD and His-CL1, or His-CL2 proteins were used for GST pulldown experiments. Four independent experiments were repeated to get similar results. **k** Diagram showing delivery of GST-NRG1-ICD or GST proteins into GABAergic interneurons in gtoGfp slices. **l** Reduced Na$^+$ current density in GABAergic interneurons treated with GST-NRG1-ICD. Representative current traces of Na$_v$ channels in GABAergic interneurons treated with recombinant GST or GST-NRG1-ICD proteins. Currents were elicited by step depolarizations from $-80$ to $+30$ mV in 5 mV increments from a holding potential of $-80$ mV. The tracts shown are for depolarizations from $-80$ to $+20$ mV. **m** I/V curves of Na$_v$ channel activation in GABAergic interneurons treated with recombinant GST or GST-NRG1-ICD proteins. **Treatment $F_{(1, 24)} = 10.89$, $P = 0.003$, two-way ANOVA, $n = 14$ cells treated with GST-NRG1-ICD, $n = 12$ cells treated with GST. Data are presented as mean values $+/-$ SEM. **n** Similar voltage-dependent activation curves of Na$_v$ channels in GABAergic interneurons treated with GST or GST-NRG1-ICD. Genotype $F_{(1, 24)} = 0.059$, $P = 0.81$, two-way ANOVA, $n = 14$ cells treated with GST-NRG1-ICD, $n = 12$ cells treated with GST. Data are presented as mean values $+/-$ SEM.

Next, we study whether NRG1-ICD plays a role in modulating Na$_v$ currents in GABAergic interneurons. SCN1A contains four transmembrane domains (I–IV) and two major cytoplasmic loops (CL1 and CL2) (Fig. 5h). To investigate whether NRG1-ICD could interact with the CL of SCN1A, we purified GST-NRG1-ICD, His-CL1, and His-CL2 proteins from bacteria (Fig. 5i). As a negative control, GST proteins did not bind with the His-CL1 or His-CL2 (Fig. 5j). The CL1 and CL2 are important for the neuromodulation and membrane localization of SCN1A, respectively[41]. No interaction between NRG1-ICD and His-CL2 (Fig. 5j) implied that NRG1-ICD might not affect the membrane localization of SCN1A. In agreement, the protein levels of SCN1A in the membrane fraction were similar between control

and gto*Nrg1* mice (Supplementary Fig. 7c). However, NRG1-ICD could bind with His-CL1 (Fig. 5j), which suggested that NRG1-ICD might modulate the function of Na$_v$1.1 channel. To further test this hypothesis, we delivered 200 nM GST-NRG1-ICD or GST proteins (as a control) into GABAergic interneurons in gtoGfp slices through a recording pipette (Fig. 5k), and 10 min afterward recorded the Na$_v$ currents using voltage clamp (Fig. 5l). The I/V curves of Na$_v$ channel activation showed a downward shift in GABAergic interneurons treated with GST-NRG1-ICD, compared with control neurons (Fig. 5m). In accord, the maximal Na$^+$ current density was significantly reduced in GABAergic interneurons treated with GST-NRG1-ICD, compared with control neurons (GST-NRG1-ICD: $-172.6 \pm 11.61$ pA/pF,

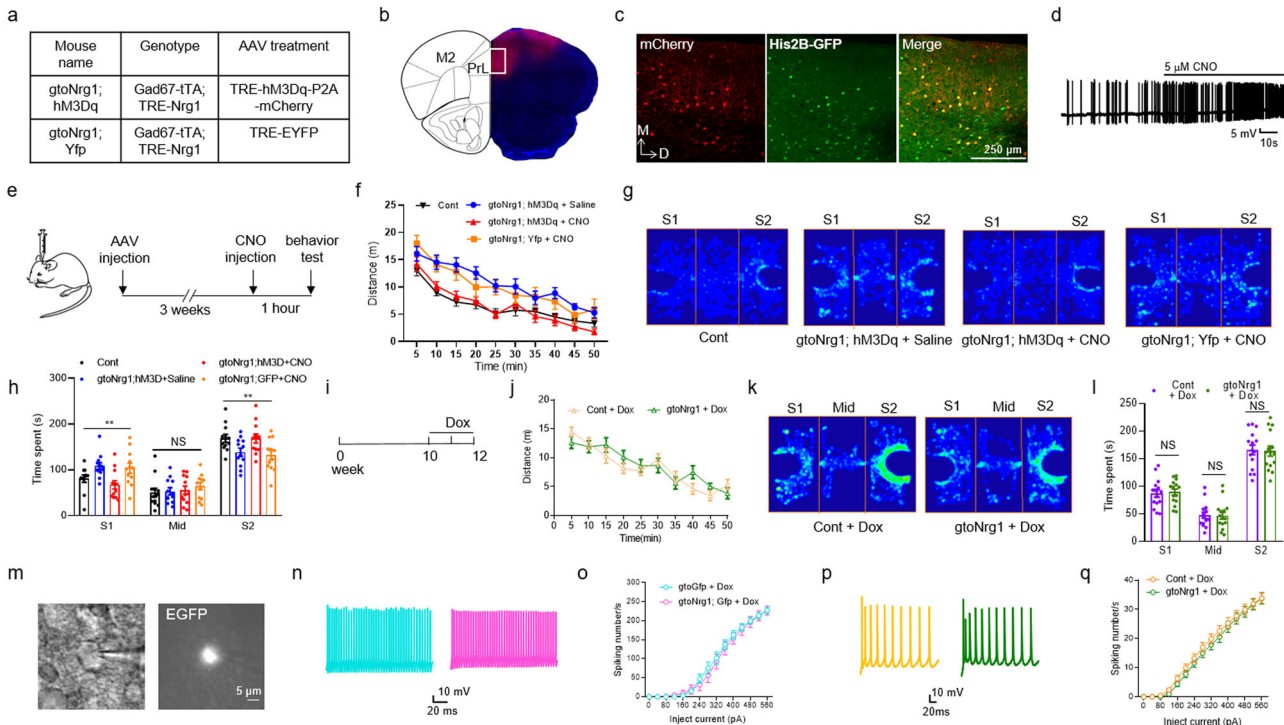

**Fig. 6 Rescue of behavioral deficits and cortical disinhibition in gtoNrg1 mice. a** The genotype and AAV treatment in different mouse lines. **b** Diagram to show the brain regions infected by AAV. **c** Expression of mCherry in GABAergic interneurons from the rectangle in panel **b**. AAV expressing TRE-hM3Dq-P2A-mCherry were injected into the PFC of gtoGfp mice. The resulting PFC slices were subjected to immunofluorescence. Three independent experiments were repeated to get similar results. Scale bar, 250 μm. **d** Increased firing of FS-GABAergic interneurons by CNO in the PFC slices of gtoNrg1; Gfp mice. The AP of FS-GABAergic interneurons were recorded before and after treatment with 5 μM CNO. **e** Experimental design. Three weeks after stereotaxic injection of AAV, the mice were received i.p. injection of CNO (5 mg/kg) 1 h before behavioral tests. **f** Rescue of hyperactivity in gtoNrg1; hM3Dq mice by CNO. Shown is travel distance in the first 50 min during the open-field test. $F_{(3, 50)} = 7.571$, $P = 0.0003$, two-way ANOVA, $n = 13$ for each group. Data are presented as mean values +/− SEM. **g** Occupancy plot of the heads from different groups of mice in the three-chamber test. S1, familiar mouse; S2, novel mouse. **h** Rescue of impaired social novelty in gtoNrg1; hM3Dq mice by CNO. Time spent in each chamber was quantified. NS not significant, **$P = 0.0068$ for S1, **$P = 0.0092$ for S2, one-way ANOVA, $n = 12$ for each group. Data are presented as mean values +/− SEM. **i** Experimental design. The 10-week-old control and gtoNrg1 mice were treated with Dox for 2 weeks before the behavioral test. **j** Similar travel distance between Dox-treated control and gtoNrg1 mice. Genotype $F_{(1, 24)} = 0.0997$, $P = 0.7549$, two-way ANOVA, $n = 13$ for each group. Data are presented as mean values +/− SEM. **k** Occupancy plot of the heads from Dox-treated control and gtoNrg1 mice in the three-chamber test. S1, familiar mouse; Mid, middle chamber; S2, novel mouse. **l** Similar social novelty between Dox-treated control and gtoNrg1 mice. Time spent in each chamber was quantified. NS, not significant, two-sided $t$ test, $n = 15$ for each group. Data are presented as mean values +/− SEM. **m** Expression of EGFP in the PFC slices from gtoNrg1; Gfp mice after treatment with Dox for 2 weeks. Four independent experiments were repeated to get similar results. Scale bar, 5 μm. **n** Representative action potential traces of FS-GABAergic interneurons in Dox-treated gtoGfp and gtoNrg1; Gfp PFC. **o** Similar I/O curves of action potential from FS-GABAergic interneurons in Dox-treated gtoGfp and gtoNrg1; Gfp PFC. Genotype $F_{(1, 28)} = 0.094$, $P = 0.762$, two-way ANOVA, $n = 15$ cells from four mice for each group. Data are presented as mean values +/− SEM. **p** Representative action potential traces of pyramidal neurons in Dox-treated gtoGfp and gtoNrg1; Gfp PFC. **q** Similar I/O curves of the action potential from pyramidal neurons in Dox-treated control and gtoNrg1 PFC. Genotype $F_{(1, 57)} = 0.433$, $P = 0.513$, two-way ANOVA, $n = 29$ cells from five gtoGfp mice, $n = 30$ cells from five gtoNrg1; Gfp mice. Data are presented as mean values +/− SEM.

$n = 14$, control: $-225.3 \pm 15.59$ pA/pF, $n = 12$, $P = 0.0109$, $t$ test). By contrast, membrane depolarizations required for half-maximal activation of Na$_v$ channels were comparable between GABAergic interneurons treated with GST-NRG1-ICD and control neurons ($V_{1/2} = -29.86 \pm 0.76$ mV for GST-NRG1-ICD-treated neurons, $n = 14$, $V_{1/2} = -28.76 \pm 0.69$ mV for control neurons, $n = 12$, $P = 0.3023$, $t$ test) (Fig. 5n). Together, these results indicated that acute application of NRG1-ICD into GABAergic interneurons attenuated the peak Na$^+$ current density but without changing the voltage dependence of Na$_v$ channel activation, which is similar to the results in gtoNrg1; Gfp mice (Fig. 5c–e).

**The causal link between cortical disinhibition and behavioral deficits.** To further demonstrate whether cortical disinhibition in PFC causes behavioral deficits in gtoNrg1 mice, we performed

rescue experiments using chemogenetic approaches. Toward this goal, we bilaterally injected adeno-associated virus (AAV) expressing TRE-hM3Dq-P2A-mCherry, an excitatory DREADD (designer receptors exclusively activated by designer drugs[42]) or TRE-EYFP (as a control) into PFC of gtoNrg1 mice. The resulting gtoNrg1 mice expressing TRE-hM3Dq-P2A-mCherry or TRE-EYFP were named gtoNrg1; hM3Dq or gtoNrg1; Yfp mice, respectively (Fig. 6a). The brain regions infected by AAV were mainly PrL and M2 motor cortex (Fig. 6b). Since the expression of hM3Dq is controlled by TRE promoter and tTA, hM3Dq is only expressed in GABAergic interneurons from gtoNrg1; hM3Dq mice (Fig. 6c). Since we used the excitatory DREADD, administration of CNO increased the excitability of FS-GABAergic interneurons in gtoNrg1; hM3Dq mice (Fig. 6d).

Three weeks after AAV administration, mice were i.p. injected with CNO or saline 1 h prior to the behavioral test (Fig. 6e).

When gto*Nrg1*; hM3Dq mice were treated with saline, they showed hyperactivity compared with controls (Fig. 6f). However, they showed normal locomotion after treatment with CNO (Fig. 6f). The effects of CNO are specific because it cannot rescue hyperactivity in gto*Nrg1*; Yfp mice (Fig. 6f). These results indicate that activation of GABAergic interneurons in the PFC could rescue hyperactivity in gto*Nrg1* mice.

When gto*Nrg1*; hM3Dq mice were treated with saline, they showed impairment in social novelty compared with controls (Fig. 6g, h). However, the social novelty became normal in gto*Nrg1*; hM3Dq mice after treatment with CNO (Fig. 6g, h). The effects of CNO are specific because it cannot reverse impaired social novelty in gto*Nrg1*; Yfp mice (Fig. 6g, h). These results indicated that activation of GABAergic interneurons in the PFC could reverse the social behavioral deficit in gto*Nrg1* mice. Altogether, these data demonstrate a causal link between PFC disinhibition and behavioral deficits in gto*Nrg1* mice.

**Dependence of behavioral deficits and cortical disinhibition on continuous NRG1 overexpression.** Since *Gad67*-tTA starts to be expressed from embryonic stage[27], the phenotypes seen in gto*Nrg1* mice could be due to overexpressing NRG1 during development or the persistent NRG1 increase in adulthood. The rescue experiments using chemogenetic approaches suggested that the behavioral deficits in gto*Nrg1* mice might be reversible. To further test this hypothesis, adult mice at 2.5-month-old were fed with Dox-containing water for 2 weeks (Fig. 6i). To eliminate possible compounding effects of Dox, both control and gto*Nrg1* mice were subjected to Dox treatment. Notably, Dox-treated gto*Nrg1* and control mice traveled a similar distance in the open field (Fig. 6j), suggesting normal locomotive activity in Dox-treated gto*Nrg1* mice. In addition, Dox-treated gto*Nrg1* and control mice showed similar performance in the three-chamber test (Fig. 6k, l), indicating normal social novelty in Dox-treated gto*Nrg1* mice.

If the cortical disinhibition is a contributing mechanism, it should be diminished when NRG1 expression returned to normal levels. To test this notion, Dox-treated mice were subjected to electrophysiological recordings. Please note that EGFP is still visible in the gto*Nrg1*; Gfp PFC slices after treatment with Dox for 2 weeks (Fig. 6m) due to the long half-life of EGFP. These observations enable us to visualize GABAergic interneurons in the gto*Nrg1*; Gfp PFC slices after turning off NRG1 overexpression. The I/O curve of AP from FS-GABAergic interneurons was similar between Dox-treated gtoGfp and gto*Nrg1*; Gfp mice (Fig. 6n, o), indicative of normal excitability. Likewise, the enhanced firing rate of PN in gto*Nrg1* PFC also returned to control levels after Dox treatment (Fig. 6p, q). Altogether, these results demonstrate that behavioral deficits and cortical disinhibition in gto*Nrg1* mice require continuous NRG1 overexpression.

**Discussion**

In this study, we first analyzed *Nrg1* gene expression in the postmortem PFC and found increased type I *Nrg1* expression in GABAergic interneurons from schizophrenia patients, compared with age- and sex-matched controls. Then we generated gto*Nrg1* mice where type I *Nrg1* was specifically overexpressed in GABAergic interneurons. Intriguingly, gto*Nrg1* mice exhibited cortical disinhibition at the cellular, synaptic, and neural network levels. We further demonstrated that cortical disinhibition led to behavioral deficits. Lastly, we illustrated that both cortical disinhibition and behavioral deficits in gto*Nrg1* mice depended on continuous NRG1 overexpressing. In sum, our results demonstrate mechanisms underlying cortical disinhibition related to

schizophrenia and raise a possibility that relevant brain disorders may benefit from intervention to restore NRG1 signaling and GABAergic function.

Chemogenetic activation of GABAergic interneurons may increase tonic inhibition, but cannot restore the temporal organization of inhibitory transmission. Therefore, the rescue locomotor and social novelty behavior by the chemogenetic manipulation suggests that these behaviors depend on tonic prefrontal inhibition, but do not require precise temporal regulation of this inhibition[43,44]. In line with this notion, the schizophrenia-relevant behavioral deficit such as hyperactivity was also found in other pharmacological models of disinhibition in the forebrain regions[45–48].

Recent studies from single-cell RNA sequencing revealed that *Nrg1* is expressed in both glutamatergic and GABAergic neurons in mouse and human PFC[16,17]. Previous studies mainly focused on the function of NRG1 in glutamatergic pyramidal neurons[49–52]. Overexpression of *Nrg1* in pyramidal neurons led to glutamatergic hypofunction through inhibiting glutamate release or impairing NMDA receptor[26,53–55]. This study showed that the excitability of FS-GABAergic interneurons was impaired in gto*Nrg1* mice where *Nrg1* was overexpressed in GABAergic interneurons. By contrast, the excitability of FS-GABAergic interneuron was not reduced in cto*Nrg1* mice where *Nrg1* was overexpressed in excitatory pyramidal neurons[26]. Recent studies indicated that NRG1 was highly expressed in PV-positive GABAergic interneurons and was important for the plasticity of the visual cortex[56,57]. Together, these results demonstrated the function of NRG1 in GABAergic interneurons.

gto*Nrg1* mice exhibited an increased theta and delta oscillations in the PFC under freely behavioral conditions. By contrast, a previous study showed that overexpression of NRG1 in pyramidal neurons enhanced gamma oscillations in hippocampal slices[50]. The discrepancy between these two studies might result from different neuronal types where NRG1 is overexpressed or the different conditions from in vivo and in vitro. Increased theta and delta oscillations are two of the consistent observations in schizophrenia patients[58]. By contrast, both increased and reduced gamma oscillation have been reported in schizophrenia patients under different conditions[59].

The NRG1 EGF domain acts on the ErbB4 receptor which is mainly expressed in GABAergic interneuron[35–37]. NRG1-ErbB4 signaling has been implicated in GABAergic circuity formation[35,60–63] and GABA transmission[38–40,64,65]. However, ErbB4 was not activated in the PFC of gto*Nrg1* mice. The NRG1 protein levels in gto*Nrg1* PFC were 1.5–2-folds higher than controls and were much lower than other *Nrg1* transgenic mouse lines[50,53], which might explain why ErbB4 was not activated in gto*Nrg1* PFC. We further showed that the NRG1 EGF domain did not cause the reduction of $Na_v$ currents in GABAergic interneurons. Indeed, the NRG1 EGF domain has been shown to increase the excitability of GABAergic interneurons through $K^+$ channels[66], albeit an opposite effect was observed in cultured hippocampal neurons[67].

On the other hand, NRG1-ICD can induce downstream signaling through protein–protein interaction and gene transcription[7,26,68–70]. The reduced excitability of FS-GABAergic interneurons in gto*Nrg1* mice might be due to the effects of NRG1-ICD. In support of this hypothesis was the observation that NRG1-ICD could interact with the cytoplasmic loop 1 of SCN1A and inhibit the peak $Na_v$ currents. The cytoplasmic loop 1 of SCN1A has been shown to interact with signaling proteins or be phosphorylated by protein kinases, which leads to the reduction of the peak $Na_v$ currents[41,71]. We speculate that the interaction of NRG1-ICD with the cytoplasmic loop 1 of SCN1A might account for the inhibition of peak $Na_v$ currents by

NRG1-ICD. Given that NRG1-ICD but not NRG1 EGF domain attenuated peak Na$_v$ currents in GABAergic interneurons, a parsimonious explanation of our finding is that the reduction of peak Na$_v$ currents in gtoNrg1 mice is due to NRG1-ICD. In sum, the data presented here demonstrate mechanisms underlying how Nrg1 dysregulation impairs brain function, which might provide insight into the pathophysiological mechanisms of schizophrenia.

## Methods

**Generation of gtoNrg1 mice**. The TRE-Nrg1 mice were generated as described in our paper[26]. Briefly, Nrg1β1a was cloned into the EcoR V site of pMM400. An HA tag was inserted between Ig and EGF domains. A Not I fragment containing the transgene was used for transgenic mouse production. Gad67-tTA mice[27] were kindly provided by Dr. Yuchio Yanagawa, Gunma University, Japan. TRE-H2B-GFP reporter mice were from Jackson laboratory (005104). The TRE-Nrg1 transgene mice and heterozygous Gad67-tTA knock-in mice were backcrossed with C57BL/6 mice for more than ten generations before cross-breeding. The resulting offspring contain four genotypes: wt, Gad67-tTA, TRE-Nrg1, and gtoNrg1. The Gad67-tTA heterozygous knock-in mice have an insertion of tTA cassette after the start codon of Gad67 gene[27], which might disrupt Gad67 gene expression. To avoid the potential knock-in effect on Gad67 gene expression, we used Gad67-tTA mice as littermate controls for gtoNrg1 mice. To eliminate the possible effects of the hormone cycle, male mice were used in all experiments. Animals were housed in rooms at 23 °C and 50% humidity in a 12 h light/dark cycle and with food and water available ad libitum. In some experiments, Dox (Sigma-Aldrich, catalog number D9891) was added to drinking water at 1 mg/ml in 2.5% sucrose. Animal experimental procedures were approved by the Institutional Animal Care and Use Committee of East China Normal University.

**RT-qPCR and single-cell RT-PCR**. The total RNA was isolated from the mouse brain and purified using Triazol (Invitrogen) and an RNAeasy mini kit (Qiagen), respectively. In total, 5 μg of total RNA was reverse transcribed using oligoT primers and SuperScript III reverse transcriptase (Invitrogen). One percent of the resulting cDNA was analyzed by qPCR in triplicates using SYBR Green/ROX (Fermentas) on Chromo 4 (Bio-Rad). The following primer pairs were used: type 1 Nrg1-F: gagtcagctgcaggctccaagc; type 1 Nrg1-R: gtgatgttggca gaggcactgtc; Gapdh-F: gtggagtcatactggaacatgtag; Gapdh-R: aatggtgaaggtcggtgtg. Levels of target mRNA levels were normalized to levels of Gapdh mRNA measured at the same time on the same reaction plate.

The PFC slices from gtoNrg1; Gfp mice were used for single-cell RT-PCR. The neurons expressing EGFP were GABAergic interneurons while the neurons with a triangle shape that do not express EGFP were excitatory pyramidal neurons. After a cell is patched, the intracellular contents are aspirated into the patch pipette and used for RT-PCR. Strict RNase-free solutions and pipettes were used for collecting single-cell RNA samples. We performed single neuron RT-PCR following the protocol in the paper[72]. The following primer pairs were used: HA-F: tgggaccagcatcgattacccat; HA-R: cgaccaccagcagggcgatac; vGat-F: tcacgacaaacccaagatcac; vGat-R: gtcttcgttctcc tcgtacag; vGlut1-F: cacagaaagcccagttcaac; vGlut1-R: catgtttagggtggaggtagc.

**Immunofluorescence**. The process of immunofluorescence analysis was performed as described by our previous studies[26]. Briefly, brain slices were permeabilized with 0.3% Triton-X 100 and 5% BSA in PBS and incubated with primary antibodies at 4 °C overnight. After washing with PBS three times, samples were incubated with Alexa Fluor-555 secondary antibodies (goat-anti-mouse, A32727; goat-anti-rabbit, A32732; 1:1000, Thermo Fisher) for 1 h at room temperature. Samples were mounted with Vectashield mounting medium (Vector Labs) and images were taken by Leica TCS SP8 confocal microscope. The following primary antibodies were used: rabbit anti-NeuN (1:500, Abcam, ab177487), mouse anti-GABA (1:1000, Invitrogen, PA5-32241), and rabbit anti-neurogranin (1:1000, R&D, MAB7947). Unbiased stereology TissueFAX Plus ST (Tissue Gnostics, Vienna, Austria)[73] was applied to count EGFP-positive and NeuN-positive cell number in brain slices.

**Western blot**. Homogenates of PFC were prepared in RIPA buffer containing 50 mM Tris–HCl, pH 7.4, 150 mM NaCl, 2 mM EDTA, 1% sodium deoxycholate, 1% SDS, 1 mM PMSF, 50 mM sodium fluoride, 1 mM sodium vanadate, 1 mM DTT, and protease inhibitors cocktails. The supernatant of homogenate was subjected to centrifugation ($25,000 \times g \times 1$ h) to obtain the membrane fraction which was dissolved by RIPA buffer containing 1 M urea. All the protein samples were boiled in 100 °C water bath for 10 min before western blot. Homogenates or membrane protein were resolved on SDS/PAGE and transferred to nitrocellulose membranes, which were incubated in the TBS buffer containing 0.1% Tween-20 and 5% milk for 1 h at room temperature before the addition of primary antibody for incubation overnight at 4 °C. After wash, the membranes were incubated with HRP-conjugated secondary antibody (goat-anti-mouse, G-21040; goat-anti-rabbit, G-21234; 1:2000, Thermo Fisher) in the same TBS buffer for 1 h at room

temperature. Immunoreactive bands were visualized by ChemiDocTM XRS + Imaging System (BIO-RAD) using enhanced chemiluminescence (Pierce) and analyzed with Image J (NIH). The following antibodies were used: rabbit anti-NRG1 (1:1000, Santa Cruz, sc-348), mouse anti-GAPDH (1:5000, Abways, ab0037), mouse anti-PSD95 (1:1000, Millipore, 2492127), rabbit anti-ErbB4 (1:1000, Cell Signaling, 111B2), rabbit anti-p-ErbB4 (1:200, Cell Signaling, Y1248) and rabbit anti-Na$_v$1.1 (1:500, Alomone Labs, ASC-001), mouse anti-GST (1:5000, ImmunoWay, B2101) and mouse anti-His (1:2000, ImmunoWay, B0401).

**Behavior analysis**. To eliminate possible effects that may be associated with the knock-in into Gad67 gene, Gad67-tTA mice were used as controls in behavioral analysis. Behavioral tests were performed in 2.5–3-months old, male mice. The investigators for behavioral tests were blind to genotypes and/or Dox administration. Both control and gtoNrg1 mice were treated with Dox to avoid possible compounding effects of Dox on behaviors. The behaviors were tested in the following order: open field, social interaction, social novelty, Y maze, and PPI. The intertest intervals are 2–3 days, except that between social interaction and novelty (30 min). The second batch of mice was used for nest building and buried food-finding test. The third batch of mice was used for behavioral tests after treatment with Dox. Mice were not tested for the same behavioral paradigms more than once, to avoid the effects of learning and memory.

*Open field*. Mice were placed in a chamber ($27.9 \times 27.9 \times 20.3$ cm) and monitored for movement for 50 min using an infrared camera placed above the box. The total distance, time in the center and margin of open field was measured by ANY-maze video tracking system (Stoelting).

*Prepulse inhibition (PPI)*. PPI tests were conducted in the SR-LAB TM Startle Response System (San Diego Instruments). The motion of mice, placed in a Plexiglas tube mounted on a plastic frame, was monitored by a piezoelectric accelerometer. Before the test, mice were allowed to habituate to the chamber, to the 70-dB background white noise for 5 min, and to the prepulse (20 ms white noise at 75, 80, or 85 dB) and auditory-evoked startle stimuli (120 dB, 20 ms). In the PPI test, mice were subjected to 12 startle trials (120 dB, 20 ms) and 12 prepulse/startle trials (20 ms white noise at 75, 80, or 85 dB at 100-ms intervals and 20 ms 120-dB startle stimulus). Different trial types were presented pseudo-randomly with each trial type presented 12 times, and no two consecutive trials were identical. Mouse movement was measured for 100 ms after the startle stimulus onset (sampling frequency 1 kHz) for 100 ms. PPI (%) was calculated according to the formula: (100 − (startle amplitude on prepulse-pulse trials/startle amplitude on pulse alone trials) × 100).

*Social interaction and novelty*. Adult male mice were tested for social behavior in a three-chamber box ($60 \times 40 \times 25$ cm). Each of the end chambers contains a clear Plexiglas cylinder. One cylinder is the "social" cylinder, which contains a stimulus mouse (adult wild-type male mice who never met test mice). The other cylinder is "non-social" cylinder, which is empty. The test mice were first placed in the center chamber and allowed to freely explore the chambers with two empty cylinders for 10 min. For the social interaction test, mice were given an additional 5 min to explore the chambers with a "social" cylinder (S1) and a "non-social" cylinder (O). Thirty min after the social interaction test, the mice were subjected to a social novelty test. The test mice were given an additional 5 min to explore the chambers with cylinders containing a familiar mouse (S1) and a novel mouse (S2). Six-week-old male mice on a C57BL/6 background were used as social opponents. Sessions were video-recorded, and time spent around the "social" cylinder and "non-social" cylinder, or the familiar cylinder and novel cylinder were analyzed by the ANY-maze video tracking system (Stoelting). The box and cylinder were cleaned with 75% ethanol and dried thoroughly after each test session.

*Y maze*. The Y-maze apparatus was shaped like a Y and had three identical arms ($25 \times 10 \times 10$ cm) placed at an angle of 120° with respect to each other. The three arms were respectively labeled A (start arm), B (old arm), and C (new arm). There are different visual cues on the wall at the end of each arm. Mice were placed at the end of one arm (A) and allowed to freely navigate A and B arm for 5 min while C arm is blocked. Ten minutes later, C arm is opened, and the mice were allowed to freely explore all three arms for an additional 4 min. When the limbs of mouse were positioned in the arm was considered to have entered an arm. The activity of the mice was recorded and analyzed by the ANY-maze video tracking system (Stoelting). The apparatus was cleaned with 75% ethanol and dried thoroughly after each test session.

*Nest building*. One cotton square ($5 \times 5$ cm) was placed in one cage with a single mouse. Twelve hours later, the nest was evaluated by a 5-point nest-rating scale[74]. Each nest was evaluated by six independent investigators who are blinded to the genotype of the mice. The score of each nest was averaged by that from six investigators.

*Buried food-finding test*. The mice were subject to food deprivation 18 h before test. The test begins by placing a mouse in a clean cage (36 cm L × 20 cm W × 18 cm H)

containing 3 cm deep of clean bedding. The subject is allowed to acclimate to the cage for 5 min. Then the mouse was transferred to an empty clean cage. We buried 2 g food pellet ~1 cm beneath the surface, in a random corner of the cage. Smooth out the surface and reintroduce the mouse to the cage. The mouse is considered to have uncovered the food when it starts to eat, usually holding the food with forepaws. The latency to find the food was recorded by the investigator.

**Recording in freely moving mice**. The microdrive containing eight tetrodes (each tetrode has four channels) was prepared as described in our previous study[75]. The mouse was anesthetized with pentobarbital sodium (40 mg/kg) before implanting the microdrive into PrL with the coordinates: anteroposterior (AP) 1.9 mm, mediolateral (ML) 0.5 mm relative to bregma. The tips of tetrodes were advanced to 1.4 mm from pia in the depth. After surgery, animals were housed in home cages to recover for 2 weeks. We used Plexon MAP system (Plexon, USA) to monitor neuronal signals. The tetrodes advanced 35 μm every other day via rotating the screw. When the tetrodes reached layers 2–3 of PrL, the recording was started. The mouse behavior was monitored simultaneously by a video. The awake active state (speed >3 cm/s) was determined by the video and real-time spectrum. The power of LFP at a different frequency (delta: 1–3 Hz, theta: 4–12 Hz, alpha: 13–15 Hz, beta: 16–30 Hz, low gamma: 30–50 Hz, high gamma: 55–90 Hz, HFO: 100–300 Hz) was analyzed through Welch's averaged periodogram with a 1024-ms nonoverlapping Hanning window (NFFT = 2048) in combination with the function of PWELCH in Matlab R2013a.

**Electrophysiology**. TRE-Nrg1 and Gad67-tTA mice showed no difference in electrophysiological studies, compared to wild-type mice. Gad67-tTA mice were used as controls in all electrophysiological studies. Mice were anesthetized by ketamine/xylazine (Sigma) and perfused transcardially for 1 min with 4 °C modified artificial cerebrospinal fluid (aCSF) containing (in mM) 250 glycerol, 2 KCl, 10 $MgSO_4$, 0.2 $CaCl_2$, 1.3 $NaH_2PO_4$, 26 $NaHCO_3$, and 10 glucose, to protect CNS neurons and maintain functional connectivity of brain slices. Mice were then decapitated and brains were quickly removed and chilled in ice-cold ACSF for an additional 1 min. Transverse mPFC slices (350 μm) were prepared using a Vibroslice (VT 1000 S; Leica) in ice-cold ACSF. Slices were then incubated in regular ASCF containing (in mM): 126 NaCl, 3 KCl, 1.25 $NaH_2PO_4$, 1.0 $MgSO_4$, 2.0 $CaCl_2$, 26 $NaHCO_3$, and 10 glucose for 30 min at 34 °C for recovery, and then at room temperature (25 ± 1 °C) for an additional 2–8 h. All solutions were saturated with 95% $O_2$/5% $CO_2$ (vol/vol). Dox (10 ng/ml) was present in the perfusate for experiments with slices from Dox-treated mice.

Whole-cell patch-clamp recordings from layers 2 to 3 PN in PrL were visualized with infrared optics using an upright microscope equipped with a ×40 water-immersion lens (BX51WI; Olympus) and infrared-sensitive CCD camera. All data were obtained with a HEKA EPC10 double patch-clamp amplifier. Data were low-pass filtered at 10 kHz and digitally sampled at 10 kHz with PatchMaster version 2 ×90.1. To record sEPSC and sIPSC, the pipettes were filled with the solution (in mM): 135 $CsCH_3SO_3$, 5 CsCl, 5 TEA-Cl, 20 HEPES, 0.4 EGTA, 2.5 Mg-ATP, 0.25 Na-GTP, and 1 QX314 (pH 7.25, 290 mOsm). Membrane potential was held at −70 mV for sEPSCs and 0 mV for sIPSCs, respectively. To record eIPSCs, the pipettes were filled with the solution (in mM): 130 $CsCH_3SO_3$, 10 CsCl, 10 HEPES, 0.2 EGTA, 1 $MgCl_2$, 4 Mg-ATP, 0.3 Na-GTP, and 5 QX314 (pH 7.25, 285 mOsm). The neurons were holding at −70 mV which were stimulated with a 100 μs current injection by a nichrome-wire electrode placed 50–100 μm from the soma of recorded neurons. To record mIPSCs, the concentration of CsCl was increased to 140 mM, $CsCH_3SO_3$ was omitted to enhance the driving force of $Cl^-$, and 1 μM TTX was added in the bath solution. To record $Na_v$ currents, $CdCl_2$ (120 μM) and CNQX (20 μM) were added to the aCSF to block $Ca^{2+}$ and AMPA receptor currents. The $Na_v$ currents were evoked with a series of 100 ms depolarizations from a holding potential of −80 mV to + 30 mV in 5 mV increments. Activation curves of $Na_v$ channels were fitted to Boltzmann relationships. In all protocols, the intersweep interval was 2 s. Spontaneous and miniature events were analyzed using Mini Analysis Program (Synaptosoft). E/I ratio was calculated by (charge$^{sEPSC}$ − charge$^{sIPSC}$)/ (charge$^{sEPSC}$ + charge$^{sIPSC}$), where charge$^{sEPSC}$ and charge$^{sIPSC}$ represent total charge of sEPSC and sIPSC, respectively.

The action potential was recorded by the current-clamp. Neurons were held at −80 mV and were injected with different currents (duration, 500 ms; increments, ±20 pA; from −200 to 580 pA; interval, 10 s). The input–output relationship was defined as the number of action potentials versus the amplitude of current injection. Input resistance was determined as the slope of the linear regression of the I–V plot for a series of hyperpolarizing pulses, where I is current amplitude and V is the steady-state voltage. The basic electrophysiological characteristics were measured for the first action potential waveform during the depolarization. The action potential threshold was calculated as the voltage corresponding to the peak of the third differential of the action potential waveform. All data were performed with Neuromatic version 3.0 (http://www.neuromatic.thinkrandom.com) which runs within Igor pro 6.7.3.2 (WaveMetrics).

**GST pulldown**. GST-tagged NRG1-ICD (amino acid 279 to 644 in rat NRG1 Iβ1a proteins) were expressed in Escherichia coli BL21 cells and were purified using Glutathione Sepharose$^{TM}$ 4 Fast Flow (GE Health) according to the manufacturer's instructions. His-tagged cytoplasmic loop (CL) 1 and 2 of SCN1A were purified using Ni–NTA agarose beads (QIAGEN) following the manufacturer's protocols. For binding assays, eluted His-Na$_v$1.1-CL proteins were incubated with immobilized GST-NRG1-ICD or GST for 4 h at 4 °C. The mixture was then washed, eluted, and subjected to western blot with anti-His and anti-GST antibodies (Abmart).

**Stereotaxic injection of AAV**. For virus injection, gtoNrg1 mice at age of 7–8 weeks were anesthetized with 1% pentobarbital sodium (100 mg/kg, i.p.) and were placed in a stereotaxic apparatus (RWD Life Science). Viruses were injected bilaterally in the PFC (PrL and M2 cortex) with the coordinates: anteroposterior (AP) 2.34 mm, mediolateral (ML) ± 0.75 mm, dorsoventral (DV) −2.00 mm relative to bregma. Each injection used 0.5 μl AAV and took 5 min. After injection, the glass pipette was left in place for 5 min in order to facilitate diffusion of the virus. The injection sites were examined at the end of the experiments, and animals with incorrect injection sites were excluded from the data analysis. Three weeks after AAV injection, mice were subjected to experiments. All surgery was conducted with an aseptic technique. The AAV expressing TRE-hM3Dq-P2A-mCherry or TRE-EYFP were generated in OBiO Technology Corp., Ltd. The CNO is purchased from MCE company (HY-17366).

**Statistical analysis**. Two-way ANOVA was used in behavioral analysis including open field, PPI, and electrophysiological studies including I/O curve of AP, PPR, and I/V curves of Na$_v$ channel activation. One-way ANOVA was used for the analysis of the data from three or more groups. Student's t test was used to compare data from two groups. Data were expressed as mean ± SEM unless otherwise indicated. The sample size justification was based on the previous studies[21,26,40]. According to the Wikipedia article on the normal distribution, about 95% of the values lie within two standard deviations. Our approach was to remove the data that were above (mean + 2*SD) and below (mean − 2*SD) before doing the statistical analysis. The example data shown were close to the overall mean. Statistically significant difference was indicated as follows: ***$P < 0.001$, **$P < 0.01$, and *$P < 0.05$. The statistical analysis was performed with the software of GraphPad Prism 8.

**Reporting summary**. Further information on research design is available in the Nature Research Reporting Summary linked to this article.

## Data availability

All data supporting the results presented herein are available from the article paper, Supplementary Information, and Source Data. The full-length images for all the gels or blots are provided in Supplementary Fig. 10. The web-links of databases GSE93577 and GSE93987 are as follows: https://www.ncbi.nlm.nih.gov/geo/query/acc.cgi?acc=GSE93577, https://www.ncbi.nlm.nih.gov/geo/query/acc.cgi?acc=GSE93987. All unique materials used are readily available from the corresponding author upon request. Source data are provided with this paper.

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

## Acknowledgements

This work was supported by grants from National Natural Science Foundation of China (No. 81471118 and 31861143033); grants from Shanghai Key Laboratory of Psychotic Disorders (No. 13dz2260500); grants from State Key Laboratory of Neuroscience. Dr. Dong-Min Yin is a NARSAD Young Investigator. We thank Dr. Yuchio Yanagawa for proving *Gad67*-tTA mice.

## Author contributions

Y.-Y.W. performed biochemical and behavioral experiments. B.Z. performed the electrophysiology recordings in brain slices. M.-M.W. and L.L. performed in vivo recordings. X.-L.Z. assisted the biochemical experiments. Y.-Y.W., B.Z., L.L., and D.-M.Y. analyzed the data. D.-M.Y. designed the experiments, supervised the work, and wrote the paper.

## Competing interests

The authors declare no competing interests.
