## [Peer Review File · Nature Communications]

REVIEWER COMMENTS

Reviewer #1 (Remarks to the Author):

The study of Wang et al, examines the relationship of NRG1 gene expression changes to schizophrenia and schizophrenia related behavior using existing gene expression datasets (GEO) and a novel mouse model of NRG1 overexpression in GABA neurons. The study is of interest, in particular given the focus on interneurons vs PNs, but the lack of focus on NRG1 isoforms makes the study difficult to interpret in the context of several other studies and established mouse models that have shown critical roles for different NRG1 isoforms and relationships to schizophrenia. For this reason, the current study represents an incremental advance to the field. Specific comments are below:

1. The entire study is focused on 'NRG1' expression changes in patients with schizophrenia (postmortem) and subsequent modelling of this in the mouse. NRG1 has over 50 splice isoforms and prior larger studies in schizophrenia PM brain have shown the importance of studying isoform families specifically, and select changes in gene expression have already been reported with regards to NRG1, types I, III and IV in schizophrenia. The current study fails to advance on any of the prior larger more carefully conducted investigations.
2. Sample size of the GEO database (36 vs 36 interneurons and 17 vs 19 PN) is very small and underpowered compared to prior postmortem brain studies of NRG1, which used much larger sample sizes.
3. Analysis of the GEO data and methods for data processing are not described. What probes in NRG1 were used to measure NRG1 expression levels? and how does this relate to the mouse model construct?
4. Lack of demographics for the PM samples. Table S1 should include postmortem interval (PMI) and pH and group means. PMI and pH and differences between groups can drastically influence gene expression observations.
5. Mouse behavior: Hyperactivity in the open field is not a phenotype thought to correspond to psychomotor agitation in patients with schizophrenia, it is a phenotype related to abnormalities of brain dopamine levels.
6. PPI: startle response to 120db prepulse is abnormal in NRG1gto mice thus it is very difficult to interpret any PPI abnormalities in this context.
7. Three chamber, social novelty data should be displayed as 'sniff time' and 'time spent' in each chamber not just time.

Reviewer #2 (Remarks to the Author):

Overall a very comprehensive study design with a number of informative follow ups resulting in data highly relevant to the field. Study design and focus as well as methods used appear appropriate. Comments to be addressed in the following.

NRG1 expression in GABAergic interneurons: was the sample cohort checked for any potential SNPs in the neuregulin 1 gene as this could affect data outcome and relate findings to a particular patient cohort (with mutant NRG1) rather than schizophrenia patients in general?

The cross-breeding (TRE-Nrg1 x Gad67-tTA) needs to be outlined in more detail including information on genetic backgrounds used for Gad67-tTA, any potential backcross generations for the resulting model etc. The choice of control model (i.e. Gad67-tTA) instead of using littermates or Dox-fed mice should be clarified as the latter two appear more appropriate control groups. Please comment on sex of mice used for the different experiments (behaviour, ephys etc) and test order as well as intertest intervals.

Statistics need to be outlined in more detail as both t-test and two-way ANOVAs were used for e.g. behavioural testing without clear rationale for the ANOVA (or was a repeated measures ANOVA used)? For PPI, it would be important to see if prepulse in itself caused a RM effect on the PPI phenotype of these mice. Figure legends should clarify what type of ANOVA result is shown etc.

Statement on intact hearing ability is confusing as the animals showed differential ASR at 120dB. The response rate for lower dB is much reduced so a floor effect might be evident here masking the hearing impairment of *gtoNrg1* mice.

Ymaze data need further analysis as it is important to not only compare % time in novel arm (result could be affected by differences in time spend in centre area) but to also run RM ANOVA to compare novel arm versus old/start arm to see that there was a clear preference for the novel arm in the control group.

Methods: Please clarify what mice were used as social opponents (age, sex, background) in the social preference test.

Reviewer #3 (Remarks to the Author):

This comprehensive study suggests a new mechanism by which increased NRG1 expression can lead to prefrontal cortical neural disinhibition. This is of general interest and clinical relevance because specific polymorphisms of the NRG1 gene have been associated with schizophrenia and neural disinhibition in the prefrontal cortex has emerged as a neuropathological feature of many neuropsychiatric disorders, including schizophrenia.

More specifically, the authors show in mice that increased expression of NRG1 in GABAergic inhibitory interneurons of the prefrontal cortex causes neural disinhibition within this region, as reflected by reduced excitability of inhibitory interneurons and reduced inhibition of pyramidal cells; this was accompanied by increased power of prefrontal local field potential oscillations at frequencies of <30 Hz. Importantly, the authors also elucidated the molecular mechanism underlying the neural disinhibition, showing that the intracellular domain of NRG1 reduces interneuron excitability through interaction with Na channels in these neurons. In line with the importance of this mechanism, rather than with a developmental role of NRG1, the authors also showed that stopping the overexpression of NRG1 in prefrontal interneurons of adult mice would reverse the prefrontal neural disinhibition (and some associated behavioural effects – locomotor hyperactivity and reduced social novelty preferences – see below).

Additional findings reported in the paper support the behavioral and clinical relevance of the prefrontal disinhibition caused by NRG1 overexpression.

First, the authors showed that overexpression of NRG1 in prefrontal inhibitory interneurons caused some behavioural changes, including increased locomotor activity, reduced prepulse inhibition, reduced social novelty preference and an impairment in what the authors refer to as 'working memory' on in a Y maze test. The increased locomotor activity and impaired social novelty preference were reversed by chemogenetic activation of prefrontal inhibitory interneurons (whereas the PPI reduction was unaffected by this manipulation and the impact of the chemogenetic manipulation on the other behaviors were not examined).

Second, using post mortem measurements in brains from patients with schizophrenia, the authors also showed that NRG1 was overexpressed in inhibitory interneurons of the prefrontal cortex, supporting the clinical relevance of the studies in mice.

Overall, I find this study to be of broad interest and I find the main findings compelling.

My specific comments mainly relate to the 'additional' findings (behavioral findings and post mortem

findings in brains from patients) and their interpretation.

Specific comments:

1) Post mortem findings in prefrontal cortex of patients with schizophrenia:

The authors suggest that NRG1 is selectively overexpressed in inhibitory interneurons, but not pyramidal neurons, from the prefrontal cortex of patients with schizophrenia. However, a comparison of Fig. 1a and b shows that, numerically, NRG1 overexpression in the patients is comparable for both neuron types, but only fails to reach statistical significance for the pyramidal neurons. The latter probably reflects substantially lower statistical power in the analysis of pyramidal neurons compared to the analysis of interneurons. More specifically, as indicated in the legend to Fig. 1 a and b, the sample sizes for the analysis of NRG1 expression in interneurons (n=36 patients with schizophrenia and n=36 control subjects) were substantially higher than for the analysis of the pyramidal neurons (n=17 patients with schizophrenia and n=19 control subjects). So, this is not a fair comparison, and based on this comparison it is not appropriate to claim that NRG1 overexpression in prefrontal neurons in schizophrenia is specific to interneurons. The authors should compare similar and appropriate sample sizes (also see my next point).

2) Sample size justification and sample selection/selection of data for illustration:

Related to my point 1), the authors should include sample size justifications (e.g., based on previous studies or based on considerations of statistical power) for ALL investigations/experiments reported in the paper, including the post mortem investigations on human brains and all investigations in the mouse model.

Moreover, the authors should also include a description of how samples were selected (randomly?) for the human post mortem studies.

Similarly, could they also clarify how the illustrative example data were chosen for the electrophysiological and behavioural experiments (e.g., 2d,e,i,j; 3b-e; 4b,g). Were this chosen randomly or to show individual findings that were close to the overall mean, so as to be representative?

3) Behavioural findings – relevance to schizophrenia and interpretation:

In my opinion, it is good that the authors include behavioural measurements to support the functional relevance of the prefrontal disinhibition. However, I have a couple of comments regarding the interpretation of the behavioral findings, which the authors should consider.

i) The Y-maze test of 'novel arm preference' or 'familiar arm recognition', with a 5-min retention delay between sample and test phases, is basically a test of spatial recognition memory. It is very different from the type of working memory that is impaired in schizophrenia, i.e. the ability to maintain information in an activated state for a few seconds. The authors should revise the paper to clarify this. I would recommend to remove reference to working memory deficits in schizophrenia (e.g., line 161 ff), as the behavioural assays included in the study do not assess the type of working memory that is impaired in schizophrenia.

ii) The authors may consider the possibility that the reduction in measures of social novelty preference and in measures of memory on the Y-maze may reflect locomotor hyperactivity. More specifically, locomotor hyperactivity may mask any difference in exploration between the novel mouse/arm and the familiar mouse/arm because of a ceiling effect due to generally high levels of exploration.

iii) Given the markedly reduced startle response in the gtoNRG1 mice (line 148, Fig. 2b), the authors should consider the possibility that the reduced PPI does not reflect an impairment in sensorimotor gating, but a floor effect (compare Swerdlow, 2000, Behav Pharmacol).

iv) In their interpretation of the chemogenetic 'rescue' experiments (line 322 ff), the authors may consider that the chemogenetic activation of prefrontal GABA neurons may increase tonic inhibition, but cannot, to my understanding, restore the temporal organisation of inhibitory transmission. Therefore, the rescue of normal locomotor and social novelty preference behavior by the chemogenetic manipulation suggests that these behaviors depend on tonic prefrontal inhibition, but do not require precise temporal regulation of this inhibition.

v) I appreciate that there may be word limits or limits on how many references to include, but, if

possible, the authors may wish to consider briefly how their findings of reduced startle/PPI and increased locomotor activity, and potentially the memory impairment, in this transgenic model of prefrontal, olfactory bulb, hippocampal and striatal disinhibition (line 124 ff., Fig. 1e) relate to related findings in pharmacological models of disinhibition in some of these forebrain regions (Japha&Koch, 1999, *Psychopharmacology*; Bast et al., 2001, *Psychopharmacology*; Enomoto et al., 2011, *Biol Psychiatry*; Pezze et al., 2014, *J Neurosci*; McGarrity et al., 2017, *Cereb Cortex*; Israelashvili et al., 2020, *Cortex*).

Point-to-point responses

Reviewer #1 (Remarks to the Author):

The study of Wang et al, examines the relationship of NRG1 gene expression changes to schizophrenia and schizophrenia related behavior using existing gene expression datasets (GEE0) and a novel mouse model of NRG1 overexpression in GABA neurons. The study is of interest, in particular given the focus on interneurons vs PNs, but the lack of focus on NRG1 isoforms makes the study difficult to interpret in the context of several other studies and established mouse models that have shown critical roles for different NRG1 isoforms and relationships to schizophrenia. For this reason, the current study represents an incremental advance to the field. Specific comments are below:

Response – We thank the reviewer for his/her positive comments that our study is of interest. We also appreciated the constructive critiques that will significantly improve the manuscript.

1. The entire study is focused on ‘NRG1’ expression changes in patients with schizophrenia (postmortem) and subsequent modelling of this in the mouse. NRG1 has over 50 splice isoforms and prior larger studies in schizophrenia PM brain have shown the importance of studying isoform families specifically, and select changes in gene expression have already been reported with regards to NRG1, types I, III and IV in schizophrenia. The current study fails to advance on any of the prior larger more carefully conducted investigations.

Response – It is a good suggestion to analyze different isoforms of Nrg1 in the postmortem brain tissue. The Nrg1 isoform we studied in the database GSE 93577 is full length type I (probe 11745036_at). In the revised ms, we performed additional analysis of type IV Nrg1 (probe 11755968_a_at). The new results indicated that type IV Nrg1 expression was not significantly altered in GABAergic interneurons from schizophrenia PFC (Fig. S1c in the revised ms). The database GSE 93577 did not provide probes specific for type III Nrg1, and thus future study is required to address whether the expression of type III Nrg1 is altered in GABAergic interneurons from schizophrenia PFC. As stated in the introduction, most previous postmortem studies analyzed gene expression changes in the total homogenate of schizophrenia brain, likely masking cell-type specific alterations due to cellular heterogeneity. The advance of this study is to address the cell-type specific alteration of Nrg1 expression in schizophrenia brain. Regarding mouse models, prior studies mainly focused on Nrg1 overexpression in pyramidal neurons, this study investigates how Nrg1 overexpression in GABAergic interneurons impairs brain function. In sum, our study has a significant advance in this field, as stated by Reviewer 2 and 3.

2. Sample size of the GEO database (36 vs 36 interneurons and 17 vs 19 PN) is very

small and underpowered compared to prior postmortem brain studies of NRG1, which used much larger sample sizes.

Response – This reviewer is correct that some of the prior postmortem studies of Nrg1 have larger sample sizes and we cited these studies in our manuscript. To increase the sample size for Nrg1 expression in PN, we analyzed another GEO database (GSE 93987) which included brain samples from 36 control and 36 schizophrenia patients (Fig. 1b in the revised ms). Since database GSE 93987 used the same postmortem PFC samples as database GSE 93577 (Table S1 in the revised ms), these two databases are suitable for comparing Nrg1 expression in GABAergic IN vs PN from schizophrenia PFC. We are sorry that we could not find any database with bigger sample size than GSE 93577 for Nrg1 expression in GABAergic IN.

3. Analysis of the GEO data and methods for data processing are not described. What probes in NRG1 were used to measure NRG1 expression levels? and how does this relate to the mouse model construct?

Response – The Nrg1 gene expression levels were normalized to that of Gapdh, as regularly done in RT-PCR analysis. The Nrg1 isoform we studied in the database GSE 93577 is full length type I (probe 11745036_at). The transgene mice also express type I Nrg1 as indicated in methods of the first submission. In the revised ms, we performed additional analysis of type IV Nrg1 (probe 11755968_a_at) whose expression was not significantly altered in GABAergic interneurons of schizophrenia PFC (Fig. S1c in the revised ms). We clarified these points in the revised ms (line 1093-1099).

4. Lack of demographics for the PM samples. Table S1 should include postmortem interval (PMI) and pH and group means. PMI and pH and differences between groups can drastically influence gene expression observations.

Response – This is a good suggestion. We added the demographics and other information such as PMI and pH for the PM samples in table S1 of the revised ms. The group means were also presented in table S1 of the revised ms. As indicated in the new table S1, the age, PMI and pH are similar between the PM samples from control and schizophrenia patients.

5. Mouse behavior: Hyperactivity in the open field is not a phenotype thought to correspond to psychomotor agitation in patients with schizophrenia, it is a phenotype related to abnormalities of brain dopamine levels.

Response – We changed the description about hyperactivity in the revised ms (line 144), as suggested.

6. PPI: startle response to 120db prepulse is abnormal in NRG1^{tg} mice thus it is

very difficult to interpret any PPI abnormalities in this context.

Response – We agree with the reviewer that reduced startle response to pulse alone cannot be unequivocally ascribed to a change in sensorimotor gating due to the “floor effect” (see also the comments from Reviewer 3). The floor effect refers to a reduction in both %PPI and PPI “difference scores”, but not in startle amplitude on prepulse + pulse trails (Swerdlow et al., 2000). To address this issue, we performed additional analysis of the startle response on prepulse + pulse trails. As shown in Fig. 2d of the revised ms, the startle magnitude on prepulse + pulse trails were significantly reduced in *gtoNrg1* mice compared with controls. These results suggest that reduced PPI in *gtoNrg1* mice may reflect deficits in sensorimotor gating rather than a floor effect. We added these data in the revised ms (line 152-160).

7. Three chamber, social novelty data should be displayed as ‘sniff time’ and ‘time spent’ in each chamber not just time.

Response – The social novelty data were displayed as “time spent in each chamber” (Fig. 2g in the revised ms), as suggested. The new data indicated that *gtoNrg1* mice spent lesser time in the chamber with novel mice (S2) but stayed longer in the chamber with familiar mice (S1), compared with controls (Fig. 2e-g in the revised ms), indicating impaired social novelty.

Reviewer #2 (Remarks to the Author):

Overall a very comprehensive study design with a number of informative follow ups resulting in data highly relevant to the field. Study design and focus as well as methods used appear appropriate. Comments to be addressed in the following.

Response – We thank the reviewer for his/her positive comments that our study is very comprehensive and highly relevant to the field. We also appreciated the constructive critiques that will significantly improve the manuscript.

NRG1 expression in GABAergic interneurons: was the sample cohort checked for any potential SNPs in the neuregulin 1 gene as this could affect data outcome and relate findings to a particular patient cohort (with mutant NRG1) rather than schizophrenia patients in general?

Response – This is a good point. Some of the prior postmortem studies linked *Nrg1* SNPs with *Nrg1* gene expression levels. However, the GEO database used in this study focused on the cell-type specific gene transcriptome and did not provide the information on gene SNPs. Future study is required to address whether the finding here is related to a particular patient cohort with mutant *Nrg1*.

The cross-breeding (TRE-Nrg1 x Gad67-tTA) needs to be outlined in more detail

including information on genetic backgrounds used for Gad67-tTA, any potential backcross generations for the resulting model etc. The choice of control model (i.e. Gad67-tTA) instead of using littermates or Dox-fed mice should be clarified as the latter two appear more appropriate control groups. Please comment on sex of mice used for the different experiments (behaviour, ephys etc) and test order as well as intertest intervals.

Response – The TRE-Nrg1 transgene mice (Yin et al., 2013) and heterozygous Gad67-tTA knockin mice (Tanaka et al., 2012) were backcrossed with C57BL/6 mice for more than 10 generations before cross-breeding. The resulting offspring contain four genotypes: wt, Gad67-tTA, TRE-Nrg1 and gtoNrg1. The Gad67-tTA heterozygous knockin mice have an insertion of tTA cassette after the start codon of Gad67 gene (Tanaka et al., 2012), which might disrupt Gad67 gene expression. To avoid the potential knockin effect on Gad67 gene expression, we used Gad67-tTA mice as littermate controls for gtoNrg1 mice. To eliminate the possible effects of hormone cycle, male mice were used in all experiments. The behaviors were tested as the following order: open field, social interaction, social novelty, Y maze and PPI. The intertest intervals are 2-3 days except that between social interaction and novelty (30 minutes). A second batch of mice were used for nest building and buried food-finding test. A third batch of mice were used for behavioral tests after treatment with Dox. We provided the information in methods of the revised ms (line 469-477, 545-549).

Statistics need to be outlined in more detail as both t-test and two-way ANOVAs were used for e.g. behavioural testing without clear rationale for the ANOVA (or was a repeated measures ANOVA used)? For PPI, it would be important to see if prepulse in itself caused a RM effect on the PPI phenotype of these mice. Figure legends should clarify what type of ANOVA result is shown etc.

Response – We clarified the statistics used for behavioral tests in figure legends of the revised ms. For PPI, we performed one-way-ANOVA analysis to determine if prepulse itself has effects on the PPI phenotype. As shown in Fig. S3f and g in the revised ms, the prepulse itself has significant impacts on the PPI phenotype.

Statement on intact hearing ability is confusing as the animals showed differential ASR at 120dB. The response rate for lower dB is much reduced so a floor effect might be evident here masking the hearing impairment of gtoNrg1 mice.

Response – We eliminate the statement on intact hearing ability in the revised ms, as suggested.

Ymaze data need further analysis as it is important to not only compare % time in novel arm (result could be affected by differences in time spend in centre area) but to also run RM ANOVA to compare novel arm versus old/start arm to see that there was a clear preference for the novel arm in the control group.

Response – We performed one-way-ANOVA analysis of Y-maze data to determine whether there was a preference for the novel arm in the control group. As shown in Fig. 21 of the revised ms, there was a preference for the novel arm in the control mice.

Methods: Please clarify what mice were used as social opponents (age, sex, background) in the social preference test.

Response – 6-week-old male mice on C57BL/6 background were used as social opponents. We clarified this point in methods of the revised ms (line 578-579).

Reviewer #3 (Remarks to the Author):

This comprehensive study suggests a new mechanism by which increased NRG1 expression can lead to prefrontal cortical neural disinhibition. This is of general interest and clinical relevance because specific polymorphisms of the NRG1 gene have been associated with schizophrenia and neural disinhibition in the prefrontal cortex has emerged as a neuropathological feature of many neuropsychiatric disorders, including schizophrenia.

More specifically, the authors show in mice that increased expression of NRG1 in GABAergic inhibitory interneurons of the prefrontal cortex causes neural disinhibition within this region, as reflected by reduced excitability of inhibitory interneurons and reduced inhibition of pyramidal cells; this was accompanied by increased power of prefrontal local field potential oscillations at frequencies of <30 Hz. Importantly, the authors also elucidated the molecular mechanism underlying the neural disinhibition, showing that the intracellular domain of NRG1 reduces interneuron excitability through interaction with Na channels in these neurons. In line with the importance of this mechanism, rather than with a developmental role of NRG1, the authors also showed that stopping the overexpression of NRG1 in prefrontal interneurons of adult mice would reverse the prefrontal neural disinhibition (and some associated behavioural effects – locomotor hyperactivity and reduced social novelty preferences – see below).

Additional findings reported in the paper support the behavioral and clinical relevance of the prefrontal disinhibition caused by NRG1 overexpression.

First, the authors showed that overexpression of NRG1 in prefrontal inhibitory interneurons caused some behavioural changes, including increased locomotor activity, reduced prepulse inhibition, reduced social novelty preference and an impairment in what the authors refer to as ‘working memory’ on in a Y maze test. The increased locomotor activity and impaired social novelty preference were reversed by chemogenetic activation of prefrontal inhibitory interneurons (whereas the PPI reduction was unaffected by this manipulation and the impact of the chemogenetic manipulation on the other behaviors were not examined).

Second, using post mortem measurements in brains from patients with schizophrenia, the authors also showed that NRG1 was overexpressed in inhibitory interneurons of the prefrontal cortex, supporting the clinical relevance of the studies in mice.

Overall, I find this study to be of broad interest and I find the main findings compelling.

My specific comments mainly relate to the 'additional' findings (behavioral findings and post mortem findings in brains from patients) and their interpretation.

Response - We thank the reviewer for his/her positive comments that our study is of broad interest and the main findings are compelling. We also appreciated the constructive critiques that will significantly improve the manuscript.

Specific comments:

1) Post mortem findings in prefrontal cortex of patients with schizophrenia:

The authors suggest that NRG1 is selectively overexpressed in inhibitory interneurons, but not pyramidal neurons, from the prefrontal cortex of patients with schizophrenia. However, a comparison of Fig. 1a and b shows that, numerically, NRG1 overexpression in the patients is comparable for both neuron types, but only fails to reach statistical significance for the pyramidal neurons. The latter probably reflects substantially lower statistical power in the analysis of pyramidal neurons compared to the analysis of interneurons. More specifically, as indicated in the legend to Fig. 1 a and b, the sample sizes for the analysis of NRG1 expression in interneurons (n=36 patients with schizophrenia and n=36 control subjects) were substantially higher than for the analysis of the pyramidal neurons (n=17 patients with schizophrenia and n=19 control subjects). So, this is not a fair comparison, and based on this comparison it is not appropriate to claim that NRG1 overexpression in prefrontal neurons in schizophrenia is specific to interneurons. The authors should compare similar and appropriate sample sizes (also see my next point).

Response – This is a good point. To increase the sample size for Nrg1 expression in pyramidal neurons, we analyzed another GEO database (GSE 93987) which included the PFC samples from 36 control and 36 schizophrenia patients. The new data indicated that Nrg1 expression was not significantly altered in PN of schizophrenia PFC (Fig. 1b in the revised ms). Since database GSE 93987 used the same PFC samples as database GSE 93577 (Table S1 in the revised ms), these two databases are suitable for comparing Nrg1 expression in GABAergic IN vs PN from schizophrenia PFC.

2) Sample size justification and sample selection/selection of data for illustration:

Related to my point 1), the authors should include sample size justifications (e.g., based on previous studies or based on considerations of statistical power) for ALL investigations/experiments reported in the paper, including the post mortem investigations on human brains and all investigations in the mouse model.

Response – The sample size justification was based on previous studies, e.g. $n > 10$ in each group for behavioral experiments, $n > 4$ in each group for electrophysiological experiments. For postmortem studies, n equal 36 for each group because we could not find any GEO database ($n > 36$) to address cell-type specific alteration of gene expression in schizophrenia PFC. We clarified the sample size in figure legends of the revised ms.

Moreover, the authors should also include a description of how samples were selected (randomly?) for the human post mortem studies.

Response – We analyzed all the samples in each GEO database, not selected, i.e. the GEO database totally have 36 samples for each group.

Similarly, could they also clarify how the illustrative example data were chosen for the electrophysiological and behavioural experiments (e.g., 2d,e,i,j; 3b-e; 4b,g). Were this chosen randomly or to show individual findings that were close to the overall mean, so as to be representative?

Response – The example data shown were close to the overall mean. We clarified this point in methods of the revised ms (line 702-703).

3) Behavioural findings – relevance to schizophrenia and interpretation:

In my opinion, it is good that the authors include behavioural measurements to support the functional relevance of the prefrontal disinhibition. However, I have a couple of comments regarding the interpretation of the behavioral findings, which the authors should consider.

Response - We thank the reviewer for the comments that it is good to include behavioral measurements. We also appreciated the constructive suggestions that will significantly improve the interpretation of the behavioral findings.

i) The Y-maze test of ‘novel arm preference’ or ‘familiar arm recognition’, with a 5-min retention delay between sample and test phases, is basically a test of spatial recognition memory. It is very different from the type of working memory that is impaired in schizophrenia, i.e. the ability to maintain information in an activated state for a few seconds. The authors should revise the paper to clarify this. I would recommend to remove reference to working memory deficits in schizophrenia (e.g., line 161 ff), as the behavioural assays included in the study do not assess the type of working memory that is impaired in schizophrenia.

Response – We removed reference to working memory deficits in schizophrenia in the revised ms, as suggested. The Y-maze data were presented as spatial recognition memory in the revised ms (line 173 and 178), as suggested.

ii) The authors may consider the possibility that the reduction in measures of social novelty preference and in measures of memory on the Y-maze may reflect locomotor hyperactivity. More specifically, locomotor hyperactivity may mask any difference in exploration between the novel mouse/arm and the familiar mouse/arm because of a ceiling effect due to generally high levels of exploration.

Response – This is a good point. To exclude potential influence of hyperactivity in the Y-maze, we analyzed the percentage of time exploring novel arm (Fig. 2n in the revised ms). This strategy of analyzing behavior was used by several previous studies (e.g. reviewed in (Wolf et al., 2016)). For social novelty test, we analyzed the time spent in each chamber, as suggested by Reviewer 1. The new data indicated that *gtoNrg1* mice spent lesser time in the chamber with novel mice (S2) but stayed longer in the chamber with familiar mice (S1), compared with controls (Fig. 2e-g in the revised ms), indicating impaired social novelty.

*iii) Given the markedly reduced startle response in the *gtoNRG1* mice (line 148, Fig. 2b), the authors should consider the possibility that the reduced PPI does not reflect an impairment in sensorimotor gating, but a floor effect (compare Swerdlow, 2000, Behav Pharmacol).*

Response – We agree with the reviewer that reduced startle response to pulse alone cannot be unequivocally ascribed to a change in sensorimotor gating due to the “floor effect” (See also the comments from Reviewer 1). The floor effect refers to a reduction in both %PPI and PPI “difference scores”, but not in startle amplitude on prepulse + pulse trails (Swerdlow et al., 2000). To address this issue, we performed additional analysis of the startle response on prepulse + pulse trails. As shown in Fig. 2d in the revised ms, the startle magnitude on prepulse + pulse trails were significantly reduced in *gtoNrg1* mice compared with controls. These results suggest that reduced PPI in *gtoNrg1* mice may reflect deficits in sensorimotor gating rather than a floor effect. We added these data in the revised ms (line 152-160).

iv) In their interpretation of the chemogenetic ‘rescue’ experiments (line 322 ff), the authors may consider that the chemogenetic activation of prefrontal GABA neurons may increase tonic inhibition, but cannot, to my understanding, restore the temporal organisation of inhibitory transmission. Therefore, the rescue of normal locomotor and social novelty preference behavior by the chemogenetic manipulation suggests that these behaviors depend on tonic prefrontal inhibition, but do not require precise temporal regulation of this inhibition.

Response – We agree with the reviewer that chemogenetic activation of GABAergic interneurons may increase tonic inhibition, but cannot restore the temporal organization of inhibitory transmission. Consistent with this notion, chemogenetic activation rescued locomotion and social novelty but not all behavioral deficits in

gtoNrg1 mice. We discussed this point in the revised ms (line 406-413).

v) *I appreciate that there may be word limits or limits on how many references to include, but, if possible, the authors may wish to consider briefly how their findings of reduced startle/PPI and increased locomotor activity, and potentially the memory impairment, in this transgenic model of prefrontal, olfactory bulb, hippocampal and striatal disinhibition (line 124 ff., Fig. 1e) relate to related findings in pharmacological models of disinhibition in some of these forebrain regions (Japha&Koch, 1999, Psychopharmacology; Bast et al., 2001, Psychopharmacology; Enomoto et al., 2011, Biol Psychiatry; Pezze et al., 2014, J Neurosci; McGarrity et al., 2017, Cereb Cortex; Israelashvili et al., 2020, Cortex).*

Response – We added the relevant references in discussion of the revised ms (line 410-413).

Reference

Swerdlow, N.R., Braff, D.L., and Geyer, M.A. (2000). Animal models of deficient sensorimotor gating: what we know, what we think we know, and what we hope to know soon. *Behav Pharmacol* 11, 185-204.

Tanaka, K.F., Matsui, K., Sasaki, T., Sano, H., Sugio, S., Fan, K., Hen, R., Nakai, J., Yanagawa, Y., Hasuwa, H., *et al.* (2012). Expanding the repertoire of optogenetically targeted cells with an enhanced gene expression system. *Cell Rep* 2, 397-406.

Wolf, A., Bauer, B., Abner, E.L., Ashkenazy-Frolinger, T., and Hartz, A.M. (2016). A Comprehensive Behavioral Test Battery to Assess Learning and Memory in 129S6/Tg2576 Mice. *PLoS One* 11, e0147733.

Yin, D.M., Chen, Y.J., Lu, Y.S., Bean, J.C., Sathyamurthy, A., Shen, C., Liu, X., Lin, T.W., Smith, C.A., Xiong, W.C., *et al.* (2013). Reversal of behavioral deficits and synaptic dysfunction in mice overexpressing neuregulin 1. *Neuron* 78, 644-657.

REVIEWERS' COMMENTS

Reviewer #1 (Remarks to the Author):

The authors have performed additional analysis of GEO datasets and clarified the model better. I have no further concerns and the manuscript is much improved.

Reviewer #2 (Remarks to the Author):

My comments have been addressed. Thanks.

Reviewer #3 (Remarks to the Author):

I thank the authors for addressing my comments on the original manuscript.

Following the authors' responses and revisions of the manuscript, I have a few additional comments, which the authors should address:

1) Sample size justification:

The authors should include the sample size justification provided in their response within the manuscript (or within the Supplementary Material). Also, in all cases where the sample size justification is based on previous studies, please include references to these studies in your justification.

2) Y-maze data and interpretation:

Thanks for clarifying within the manuscript that the assay measures spatial recognition memory, rather than working memory. Please also replace the reference to 'working memory' in the Abstract (line 29).

3) No evidence for reduced sensorimotor gating by NRG1 overexpression:

The additional analysis of startle data on the pre-pulse + pulse trials (Fig. 2d) clearly shows that both genotypes similarly reduce their startle response to the startle pulse with increasing intensity of the pre-pulse. The outcomes of the two-way ANOVA using genotype and prepulse intensity as independent variables are incompletely reported, but the data suggest that, in addition to the main effect of genotype (which is reported in the figure legend), the main effect of prepulse intensity may be significant, but – importantly – the data do not indicate that there is a significant interaction prepulse X genotype. The latter means that there is NO difference in the extent to which the prepulse gates the startle response to the loud pulse, so there is NO evidence for genotype to affect sensorimotor gating. The complete outcomes of the ANOVA need to be included in the manuscript and, based on these outcomes, it needs to be clearly indicated that the data only support that NRG1 overexpression reduces the startle response, but does NOT affect the ability of prepulses to 'gate', i.e. reduce, the startle amplitude. In other words, the data show that NRG1 overexpression does NOT impair sensorimotor gating, as measured using the PPI paradigm. This also needs to be clarified in the Abstract.

4) Line 410-413: 'In line with this notion, the schizophrenia-relevant behavioral deficits such as hyperactivity and impaired social novelty were also found in other pharmacological models of disinhibition in the forebrain regions 46-49.'

The references support that prefrontal, hippocampal and ventral striatal disinhibition cause locomotor hyperactivity, but do not seem to report impaired social novelty. Please correct the statement accordingly.

Point-to-point responses

Reviewer #1 (Remarks to the Author):

The authors have performed additional analysis of GEO datasets and clarified the model better. I have no further concerns and the manuscript is much improved.

Response – We thank this reviewer for his/her positive comments.

Reviewer #2 (Remarks to the Author):

My comments have been addressed. Thanks.

Response – We thank this reviewer for his/her positive comments.

Reviewer #3 (Remarks to the Author):

I thank the authors for addressing my comments on the original manuscript.

Following the authors' responses and revisions of the manuscript, I have a few additional comments, which the authors should address:

1) Sample size justification:

The authors should include the sample size justification provided in their response within the manuscript (or within the Supplementary Material). Also, in all cases where the sample size justification is based on previous studies, please include references to these studies in your justification.

Response – We included the sample size justification in the Methods section of the revised ms and added the relevant references, as suggested (line 698-699).

2) Y-maze data and interpretation:

Thanks for clarifying within the manuscript that the assay measures spatial recognition memory, rather than working memory. Please also replace the reference to 'working memory' in the Abstract (line 29).

Response – We removed “working memory” in the abstract, as suggested.

3) No evidence for reduced sensorimotor gating by NRG1 overexpression:

The additional analysis of startle data on the pre-pulse + pulse trials (Fig. 2d) clearly shows that both genotypes similarly reduce their startle response to the startle pulse with increasing intensity of the pre-pulse. The outcomes of the two-way ANOVA using genotype and prepulse intensity as independent variables are incompletely reported,

but the data suggest that, in addition to the main effect of genotype (which is reported in the figure legend), the main effect of prepulse intensity may be significant, but – importantly – the data do not indicate that there is a significant interaction prepulse X genotype. The latter means that there is NO difference in the extent to which the prepulse gates the startle response to the loud pulse, so there is NO evidence for genotype to affect sensorimotor gating.

The complete outcomes of the ANOVA need to be included in the manuscript and, based on these outcomes, it needs to be clearly indicated that the data only support that NRG1 overexpression reduces the startle response, but does NOT affect the ability of prepulses to ‘gate’, i.e. reduce, the startle amplitude. In other words, the data show that NRG1 overexpression does NOT impair sensorimotor gating, as measured using the PPI paradigm. This also needs to be clarified in the Abstract.

Response – We reported the two-way-ANOVA analysis on the effects of prepulse intensity and interaction (prepulse x genotype) in Fig. 2d. Since the p value for the interaction between prepulse and genotype is 0.0486, very close to 0.05, we agree with this reviewer that Nrg1 overexpression has minor effects on sensorimotor gating. We clarified these points in the revised ms (line 145-152 and 971-974).

4) Line 410-413: ‘In line with this notion, the schizophrenia-relevant behavioral deficits such as hyperactivity and impaired social novelty were also found in other pharmacological models of disinhibition in the forebrain regions 46-49.’

The references support that prefrontal, hippocampal and ventral striatal disinhibition cause locomotor hyperactivity, but do not seem to report impaired social novelty. Please correct the statement accordingly.

Response – We corrected the statement in the revised ms, as suggested (line 403).